# GRASSMANNIAN GEOMETRY MEETS DYNAMIC MODE DECOMPOSITION IN DMD-GEN: A NEW METRIC FOR MODE COLLAPSE IN TIME SERIES GENERATIVE MODELS

## ABSTRACT

Generative models like Generative Adversarial Networks (GANs) and Variational Autoencoders (VAEs) often fail to capture the full diversity of their training data, leading to mode collapse. While this issue is well-explored in image generation, it remains underinvestigated for time series data. We introduce a new definition of mode collapse specific to time series and propose a novel metric, DMD-GEN, to quantify its severity. Our metric utilizes Dynamic Mode Decomposition (DMD), a data-driven technique for identifying coherent spatiotemporal patterns, and employs Optimal Transport between DMD eigenvectors to assess discrepancies between the underlying dynamics of the original and generated data. This approach not only quantifies the preservation of essential dynamic characteristics but also provides interpretability by pinpointing which modes have collapsed. We validate DMD-GEN on both synthetic and real-world datasets using various generative models, including TimeGAN, TimeVAE, and DiffusionTS. The results demonstrate that DMD-GEN correlates well with traditional evaluation metrics for static data while offering the advantage of applicability to dynamic data. This work offers for the first time a definition of mode collapse for time series, improving understanding, and forming the basis of our tool for assessing and improving generative models in the time series domain.

## 1 INTRODUCTION

Generative models gained significant attention in recent years, driven by recent advancements in computational power, the availability of extensive datasets, and breakthrough developments in machine learning (ML) algorithms. Notably, models like Generative Adversarial Networks (GANs) and Variational Autoencoders (VAEs) excel at learning compact representations of data Goodfellow et al. (2014); Pu et al. (2016); Oublal et al. (2024). These models are invaluable for generating realistic samples for testing, training other models, or augmenting datasets to enhance the performance of ML algorithms Zheng et al. (2023); Abdollahzadeh et al. (2023). However, recent studies have revealed that generative models sometimes fail to produce diverse samples, leading to reduced effectiveness in applications that necessitate a broad spectrum of variations Aboussalah et al. (2023); Berns (2022); New et al. (2023). An illustration of this challenge can be seen in GANs, which frequently experience mode collapse, a phenomenon where the generator focuses on a limited subset of the data distribution, leading to the production of repetitive or similar samples rather than capturing the full diversity of the training data Bang & Shim (2018); Pan et al. (2022); Eide et al. (2020). VAEs also face a phenomenon called posterior collapse, where the model tends to generate outputs that are similar or indistinguishable for different inputs. This limitation reduces the model's ability to produce diverse samples He et al. (2019); Wang et al. (2021). Diffusion models, while generally robust against mode collapse compared to GANs and VAEs, are not entirely immune to difficulties in covering the full data distribution. These challenges can become apparent when high classifier-free guidance is employed during sampling or when training on small datasets Ho & Salimans (2022); Sadat et al. (2024); Qin et al. (2023).

The issue of diversity in generative models has received significant attention in fields such as computer vision and natural language processing Lee et al. (2023); Chung et al. (2023); Liu et al. (2020); Ibarrola & Grace (2024), however, it remains relatively underexplored in the domain of time series data. Due to the inherent time-varying nature of time series, the traditional definition of mode collapse is inadequate, necessitating a new framework for this context. Unlike images, time series data often exhibits patterns that change over time, such as trends, seasonality, or cyclic behavior. Moreover, in the context of mode collapse for generative models, particularly GANs, modes refer to distinct patterns or sub-groups within the data distribution that the model is attempting to learn and replicate. However, defining modes for time series data is challenging due to several intrinsic characteristics of time series as they must be able to capture these evolving patterns rather than generating repetitive or static sequences.

**Contributions.** The contributions of our work are as follows:

- **New Definition of Mode Collapse for Time Series:** We introduce a new definition of mode collapse specifically for time series data, leveraging DMD to capture and analyze coherent dynamic patterns.

- **Development of DMD-GEN Metric:** We propose DMD-GEN, a new metric to detect mode collapse, which consistently aligns with traditional generative model evaluation metrics while offering unique insights into time series dynamics.

- **Enhanced Interpretability:** The DMD-GEN metric provides increased interpretability by decomposing the underlying dynamics into distinct modes, allowing for a clearer understanding of the preservation of essential time series characteristics.

- **Efficiency in Time Complexity:** Our approach offers significant computational efficiency as it requires no additional training, making it highly scalable for real-time applications.

## 2 BACKGROUND AND RELATED WORK

### 2.1 MODE COLLAPSE FOR TIME SERIES

Before delving into the details of our new evaluation metric incorporating the aforementioned concepts, it is worth highlighting the challenges to be addressed in order to measure mode collapse when dealing with time series. *(i) Capturing Modes:* For time series, we need to consider *modes* that represent different unfolding patterns over time. Real-world time series data rarely exhibits a single and clean pattern Lim & Zohren (2021); Kantz & Schreiber (2004). Instead, time series data often exhibits multiple patterns simultaneously, e.g. superimposed on long-term trends and shorter-term fluctuations representing unfolding modes. This makes it challenging to isolate and identify the specific mode of interest. Moreover, unlike data with clear separations like images or text, time series data is continuous. This makes it challenging to pinpoint the exact start and end points of a specific unfolding pattern (mode). *(ii) Similarity Measurement:* Time series data often have different characteristics that make standard distance metrics such as Euclidean distance less effective as a similarity measure. For example, Euclidean distance is sensitive to different dimension scales in each dimension, and when the dimensionality is high, the distances may become dominated by differences in certain dimensions, leading to inaccuracies in similarity measurement. In addition, Euclidean distance does not take into account the temporal dependencies present in time series data. It treats each timestamp as independent, which may not be appropriate for time series data where the ordering and temporal relationships between data points are crucial. More sophisticated measures, such as Dynamic Time Warping (DTW) Müller (2007), are designed to handle these temporal dependencies by aligning sequences in a way that minimizes distance, however, it cannot effectively capture the underlying modes or coherent dynamic patterns in the data. This inability to recognize and preserve the essential modes means DTW falls short in assessing mode collapse.

Unlike in images, the mode collapse issue in time series cannot be easily distinguished with human eyes. Therefore, this area remains relatively under-explored. Few studies have formally addressed this problem and proposed solutions within the domain of time series. The work of Lin et al. (2020) developed a custom auto-normalization heuristic which normalizes each time series individually rather than normalizing over the entire dataset. However, the custom auto-normalization heuristic only concerns mode collapse defined by the different offset values between each time series while it

does not take into account whether the model has generated the entire dataset's trends and seasonality. Additionally, *DC-GAN* Min et al. (2023) is the first time-series GAN that can generate all the temporal features in a multimodal distributed time series. DC-GAN relies on the concept of directed chain stochastic differential equations (DC-SDEs) Detering et al. (2020). While the authors did not explicitly mention mode collapse, it would be valuable to investigate further whether the DC-GAN successfully captures the full range of trends and seasonality present in the dataset.

## 2.2 DYNAMIC MODE DECOMPOSITION

*Dynamic Mode Decomposition* (DMD) Schmid (2010); Peters (2019) is a data-driven and model-free method used for analyzing the underlying dynamics of complex systems such as fluid dynamics. It is used to extract modal descriptions of a nonlinear dynamical system from data without any prior knowledge of the system required. DMD yields direct information concerning the data dynamics, allowing us to compare multiple time series. Given a dynamical system $\dot{\mathbf{x}}(t) = \mathbf{f}(\mathbf{x}(t), t; \mu)$, where $\mathbf{x}(t) \in \mathbb{R}^n$ is a vector of dimension $n$ representing the state of the dynamical system at time t, $\mu \in \mathbb{R}^p$ contains parameters of the system, and $\mathbf{f} : \mathbb{R}^n \times \mathbb{R} \times \mathbb{R}^p \to \mathbb{R}^n$ represents the dynamics. DMD approximates a locally linear dynamical system $\dot{\mathbf{x}} \approx \mathcal{A}\mathbf{x}$, where the operator $\mathcal{A}$ is the best-fit linear approximation to $\mathbf{f}$ through regression. This linear approximation allows the representation of the system's behavior in a simplified framework and helps construct reduced-order models that capture the essential dynamics of the systems. This is particularly useful for systems with large state spaces like fluid dynamics Kutz (2017); Jiaqing & Weiwei (2018). Analogously, we approximate the dynamical system linearly for discrete time series. Given a dynamical system $\mathbf{x} : t \in \mathbb{R} \mapsto \mathbf{x}(t) \in \mathbb{R}^n$, we generate discrete-time snapshots of length $m$, arranged into two data matrices $\mathbf{X}, \mathbf{X}' \in \mathbb{R}^{n \times m}$ defined as follows,

$$\mathbf{X} = \left[ \begin{array}{cccc} | & | & & | \\ \mathbf{x}_0 & \mathbf{x}_1 & \cdots & \mathbf{x}_{m-1} \\ | & | & & | \end{array} \right], \quad \mathbf{X}' = \left[ \begin{array}{cccc} | & | & & | \\ \mathbf{x}_1 & \mathbf{x}_2 & \cdots & \mathbf{x}_m \\ | & | & & | \end{array} \right].$$

These snapshots are taken with a time-step $\Delta t$ small enough to capture the highest frequencies in the system's dynamics, i.e., $\forall k \in \mathbb{N}, \mathbf{x}_k = \mathbf{x}(k\Delta t)$. Assuming uniform sampling in time, we approximate the dynamical system linearly as $\mathbf{x}_{k+1} \approx \mathbf{A}^\star \mathbf{x}_k$, where $\mathbf{A}^\star \in \mathbb{R}^{n \times n}$ is the best-fit operator, i.e., $\mathbf{A}^\star = \arg\min_{\mathbf{A}} \|\mathbf{X}' - \mathbf{A}\mathbf{X}\|_F = \mathbf{X}'\mathbf{X}^\dagger$, where $\|.\|_F$ is the Frobenius norm and $\mathbf{X}^\dagger$ is the Moore-Penrose generalized inverse of $\mathbf{X}$. The optimal operator $\mathbf{A}^\star$ is linked with the operator $\mathcal{A}$, defined earlier, by the equation $\mathbf{A}^\star = \exp(\mathcal{A}\Delta t)$, cf. Appendix A.

The DMD operator $\mathbf{A}^\star$ is intricately connected to the *Koopman theory* Mezić (2013); Brunton et al. (2021). The DMD operator acts as an approximate representation of the Koopman operator within a finite-dimensional subspace of linear measurements, The connection was originally established by Rowley et al. (2009). Koopman operators, provide a powerful framework for globally linearizing nonlinear dynamical systems, offering valuable insights into their dynamics through spectral analysis Colbrook & Townsend (2024). By examining the eigenvalues $\mathbf{\Lambda} = diag(\lambda_1, \ldots, \lambda_r)$ and the corresponding eigenvectors $\mathbf{\Phi} = \|_{s=1}^r \phi_s \in \mathbb{R}^{n \times r}$ of the DMD operator $\mathbf{A}^\star$, where $r$ is the rank of the matrix $\mathbf{X}$ and $\|$ is the concatenation operator, we can effectively capture and understand the underlying patterns and dynamics of the system. Conventionally, the eigenvectors and their corresponding eigenvalues are arranged in descending order based on the magnitude of the eigenvalues, i.e., $|\lambda_1| \geq \ldots \geq |\lambda_r|$.

For a high-dimensional state vector $\mathbf{x} \in \mathbb{R}^n$, the matrix $\mathbf{A}^\star$ comprises $n^2$ elements, making its representation and spectral decomposition computationally challenging. To address this, we apply dimensionality reduction to efficiently compute the dominant eigenvalues and eigenvectors of $\mathbf{A}^\star$ by constructing a reduced-order approximation $\tilde{\mathbf{A}} \in \mathbb{R}^{r \times r}$. The DMD approximation at each time step $k = 0, 1, \ldots, m$ can be expressed as follows,

$$\forall k, \quad \mathbf{x_k} = \sum_{j=1}^r \phi_j \lambda_j^k b_j = \mathbf{\Phi}\mathbf{\Lambda^k}\mathbf{b}, \tag{1}$$

where $\phi_j$ are DMD modes (eigenvectors of the A matrix, $\lambda_j$ are DMD eigenvalues (eigenvalues of the A matrix), and $b_j$ is the mode amplitude ($\mathbf{b} = \mathbf{\Phi}^\dagger \mathbf{x_0}$ in the matrix notation). The detailed steps for this process are provided in Appendix A.3. According to the Equation 1, DMD modes can also

be viewed as bases that span a subspace representing coherence patterns among the attributes of $\mathbf{x}(t)$. DMD decomposes a complex time series into a collection of simpler, coherent modes. Each mode captures a specific aspect of the system's behavior, such as an oscillation, an exponential growth/decay, or a traveling wave. The modes are spatial fields that often identify coherent structures in the flow, which are captured by the DMD eigenvalues $\mathbf{\Lambda}$ and eigenvectors $\mathbf{\Phi}$. For example, the imaginary part of the eigenvalues $\mathbf{\Lambda}$ determines the oscillation frequency, while the real part indicates the rate of decay Tu (2013); Chen et al. (2012). The DMD eigenvectors capture the spatial structure or spatial coherence of a particular mode. Therefore, by examining the components of the eigenvectors, we can know which parts of the system contribute to the overall dynamic behavior represented by the corresponding eigenvalue. Consequently, DMD provides a rigorous framework for defining and analyzing mode collapse in time series, offering a powerful role in understanding the dynamics of a time series.

## 3 DMD-GEN: TOWARD AN EXPLAINABLE METRIC FOR CAPTURING MODE COLLASPE IN TIME SERIES

### 3.1 NOTATIONS

We explore generative models, denoted as $\mathcal{G}$, such as GAN, VAE, or the Diffusion Model, specifically adapted for time series data, as discussed in prior works Yoon et al. (2019); Yuan & Qiao (2024). These models are trained on a dataset comprising $N$ time series, each with a fixed length, represented as $\{\mathbf{X}_i\}_{i=1}^N$. During the inference phase, we utilize these trained models to synthetically generate a set of $\widetilde{N}$ time series, denoted as $\{\widetilde{\mathbf{X}}_j\}_{j=1}^{\widetilde{N}}$. Both the original and generated time series are assumed to have a consistent length (number of time points), denoted as $\ell$, and dimensionality (number of features), represented as $n$. Formally, for any pair of indices $i$ and $j$, the original and generated time series $\mathbf{X}_i$ and $\widetilde{\mathbf{X}}_j$ are elements of the Euclidean space $\mathbb{R}^{n \times \ell}$.

### 3.2 MEASURING THE SIMILARITY BETWEEN TIME SERIES USING THEIR RESPECTIVE DMD MODES

We are interested in measuring the dynamics similarity between a real time series $\mathbf{X}_i$ and a generated time series $\widetilde{\mathbf{X}}_j$. Comparing the respective DMD eigenvectors offers a valuable approach to assess similarity between time series, as they allow for the recognition of dynamic patterns. According to Equation 1, the dominant modes can be captured using the eigenvectors corresponding the largest DMD eigenvalues. If both time series exhibit similar dominant modes (eigenvectors with high eigenvalues), it suggests they share similar underlying dynamics. This could be the case for seasonal patterns in temperature data or business cycles in economic data.

**Definition 1** (**Temporal Modes**). Given a time series $\mathbf{X} = [\mathbf{x}_1, \ldots, \mathbf{x}_\ell] \in \mathbb{R}^{\ell \times n}$, we define the corresponding set of temporal modes $\mathcal{M}_k(\mathbf{X})$ as the set of eigenvectors $\{\phi_1, \ldots, \phi_k\}$ associated with the $k-$largest eigenvalues of the corresponding DMD operator, which is the operator that governs the evolution of the system state generating the time series data. Mathematically, we represent $\mathcal{M}_k(\mathbf{X})$ as a matrix formed by the concatenation of the eigenvectors, i.e.,

$$\mathcal{M}_k(\mathbf{X}) = \overset{k}{\underset{s=1}{\|}} \phi_s = \left[ \begin{array}{cccc} | & | & & | \\ \phi_1 & \phi_2 & \cdots & \phi_k \\ | & | & & | \end{array} \right]^\top \in \mathbb{R}^{k \times n}.$$

Definition 1 captures the essence of a mode in terms of the dominant eigenvalues and eigenvectors of the DMD operator, highlighting the significant dynamic structures in the time series data. Therefore, to compare the similarity between two time series, such as an original time series $\mathbf{x}$ and another $\widetilde{\mathbf{x}}$ generated by a generative model $\mathcal{G}$, we can focus on the similarity of their respective temporal modes $\mathcal{M}_k(\mathbf{X})$ and $\mathcal{M}_k(\widetilde{\mathbf{X}})$. These modes, as defined using the $k-$largest eigenvectors and eigenvalues from DMD, represent the key dynamic patterns in the time series. Comparing these modes allows us to evaluate how well the generative model has preserved the essential dynamics of the original data.

However, comparing the distances of eigenvectors $\mathcal{M}_k(\mathbf{X})$ and $\mathcal{M}_k(\widetilde{\mathbf{X}})$ is mathematically challenging. Although these vectors have similar dimension, the eigenvector subspaces are not necessarily

aligned and usually have different bases which makes the comparison not straightforward. As the eigenvector subspaces have both the same dimension, we can use the concept of Grassmann manifold to compare the similarity between $\mathcal{M}_k(\mathbf{X})$ and $\mathcal{M}_k(\widetilde{\mathbf{X}})$ Kobayashi & Nomizu (1996). A Grassmannian manifold $\mathrm{Gr}(k, n)$ is the space of all $k$-dimensional linear subspaces of an $n$-dimensional vector space $\mathbb{R}^n$ or $\mathbb{C}^n$. Formally, the Grassmannian can be defined as follows:

**Definition 2** (**Grassmannian manifold**). Let $V$ be an $n$-dimensional vector space over a field $\mathbb{F}$ (typically $\mathbb{R}$ or $\mathbb{C}$). The Grassmannian manifold $\mathrm{Gr}(k, n)$ is the set of all $k$-dimensional subspaces of $V$, where $1 \leq k \leq n$. Mathematically, it can be expressed as:

$$\mathrm{Gr}(k, n) = \{W \subseteq V : \dim(W) = k\}.$$

The Riemannian distance between two subspaces is the length of the shortest geodesic connecting the two points on the Grassmann manifold, which can be calculated based on *principal angles* between subspaces, which in our case represent the temporal modes, c.f. See Definition 3.

**Definition 3** (**Principal Angles Between Temporal Modes**). Let the columns of $\mathcal{M}_k(\mathbf{X})$ and $\mathcal{M}_k(\widetilde{\mathbf{X}})$ represent two linear subspaces $\mathbf{U}$ and $\widetilde{\mathbf{U}}$, respectively. The principal angles $0 \leq \theta_1 \leq \cdots \leq \theta_r \leq \pi/2$ between the two subspaces are defined recursively as follows:

$$\cos\theta_k = \max_{u \in \mathbf{U}} \ \max_{v \in \widetilde{\mathbf{U}}} u^\top v \quad s.t. \ \left\{ \begin{array}{l} u^\top u = v^\top v = 1 \\ u^\top u_i = v^\top v_i = 0, i = 1, \ldots, k-1 \end{array} \right. \tag{2}$$

The work of Björck & Golub (1973) have shown that the principal angles can be efficiently computed via the singular value decomposition (SVD) of $\mathbf{Q}^\top \widetilde{\mathbf{Q}}$, where $\mathbf{QR}$ and $\widetilde{\mathbf{Q}}\widetilde{\mathbf{R}}$ are the QR factorizations of $\mathcal{M}_k(\mathbf{X})$ and $\mathcal{M}_k(\widetilde{\mathbf{X}})$, respectively. Writing the SVD decomposition of $\mathbf{Q}^\top \widetilde{\mathbf{Q}}$ as $\mathbf{Q}^\top \widetilde{\mathbf{Q}} = \mathbf{U_{ang}}\mathbf{\Sigma_{ang}}\mathbf{V_{ang}^\top}$ where $\mathbf{\Sigma_{ang}}$ is a diagonal matrix. If $s$ is the rank of $\mathbf{\Sigma_{ang}}$, then the principal angles corresponds to the *arcos* of the $s$ first singular values of $\mathbf{\Sigma_{ang}}$, i.e. $\mathbf{\Theta} = diag(\cos^{-1}\sigma_1, \ldots, \cos^{-1}\sigma_s)$. In Bito et al. (2019), they incorporated the following Grassmann metrics into several learning algorithms. Such metrics include the *Projection* distance defined by the Frobenius norm of the matrix $sin\mathbf{\Theta}$, i.e.

$$d_P\left(\mathcal{M}_k(\mathbf{X}), \mathcal{M}_k(\widetilde{\mathbf{X}})\right) = \|\mathbf{sin}(\mathbf{\Theta})\|_F = \left(\sum_{k=1}^{s} \sin^2\theta_k\right)^{1/2}. \tag{3}$$

The new spectral distance based on the principal angles between the subspaces spanned by the normalized eigenvectors can serve as a similarity metric. The smaller the principal angles, the closer the subspaces are to each other, indicating a higher degree of similarity.

Given two sets of temporal modes $\mathcal{M}_k(\mathbf{X})$ and $\mathcal{M}_k(\widetilde{\mathbf{X}})$, it is known that there are many geodesics linking the two points on the $\mathrm{Gr}(k, n)$ Wong (1967); Sun et al. (2016). If all the principal angles are between 0 and $\pi/2$, it is proven that the geodesic is unique Wong (1967); Sun et al. (2016).

### 3.3 MEASURING MODE COLLAPSE FOR TIME SERIES

To measure the mode collapse of generative models for time series data, we propose a novel approach using Optimal Transport (OT) to evaluate the similarity and preservation of modes between real and generated time series. Initially, DMD is employed to extract key modes from both real and generated time series, capturing the significant dynamic patterns inherent in each dataset. For a given $L$ sampled batchs of real time series $\mathcal{X} = \{\mathbf{X}_1, \mathbf{X}_2, \ldots, \mathbf{X}_L\}$ and generated time series $\widetilde{\mathcal{X}} = \{\widetilde{\mathbf{X}}_1, \widetilde{\mathbf{X}}_2, \ldots, \widetilde{\mathbf{X}}_L\}$, we compute the respective sets of DMD modes, $\{\mathcal{M}_k(\mathbf{X_i})\}_{i=1}^{L}, \{\mathcal{M}_k(\widetilde{\mathbf{X_j}})\}_{j=1}^{L}$, which encapsulate the dominant temporal dynamics. We then construct a cost matrix $\mathbf{C}$, where each element $\mathbf{C}_{ij}$ quantifies the dissimilarity between modes $\mathcal{M}_k(\mathbf{X_i})$ and $\mathcal{M}_k(\widetilde{\mathbf{X_j}})$, using principal angle based metric. The OT problem is solved to find an optimal transport plan $\gamma^\star$ that minimizes the total transportation cost, thus identifying the best mapping between the modes of the real and generated time series using the Wasserstein distance defined as follows,

$$\mathbb{E}_{i,j}\left[W_p\left(\mathcal{M}_k(\mathbf{X_i}), \mathcal{M}_k(\widetilde{\mathbf{X_j}})\right)\right] = \mathbb{E}_{i,j}\left[\min_{\gamma \in \Pi}\langle\gamma, \mathbf{C}\rangle_p\right] = \mathbb{E}_{i,j}\left[\left(\min_{\gamma \in \Pi}\sum_{i,j=1}^{L}\gamma_{ij}\mathbf{C_{ij}}^p\right)^{\frac{1}{p}}\right], \tag{4}$$

where $p$ is the order of Wasserstein distance and $\Pi$ is the set of all joint probability distributions. The resulting Wasserstein distance, derived from the optimal transport plan, serves as a robust measure of mode collapse: a lower Wasserstein distance indicates better preservation of the original modes in the generated data, highlighting the effectiveness of the generative model in maintaining the intrinsic dynamic patterns of the time series. The geodesic distance $\gamma$ in Equation 4 is defined by Theorem 4.

**Theorem 4** (DMD Mode Geodesic). *Let $\mathcal{M}_k(\mathbf{X}), \mathcal{M}_k(\widetilde{\mathbf{X}}) \in \mathbb{R}^{n \times k}$ be matrices whose columns form orthonormal bases of two $k$-dimensional subspaces of $\mathbb{R}^n$. Let $\Theta = \operatorname{diag}(\theta_1, \theta_2, \ldots, \theta_k)$ be the diagonal matrix of principal angles between the subspaces spanned by $\mathcal{M}_k(\mathbf{X})$ and $\mathcal{M}_k(\widetilde{\mathbf{X}})$. Let $\Delta \in \mathbb{R}^{n \times k}$ be an orthonormal matrix such that*

$$\mathcal{M}_k(\widetilde{\mathbf{X}}) = \mathcal{M}_k(\mathbf{X}) \cos(\Theta) + \Delta \sin(\Theta). \tag{5}$$

*Then, the geodesic linking $\mathcal{M}_k(\mathbf{X})$ and $\mathcal{M}_k(\widetilde{\mathbf{X}})$ on the Grassmann manifold $\operatorname{Gr}(k, n)$ is given by*

$$\gamma(t) = \mathcal{M}_k(\mathbf{X}) \cos(t\Theta) + \Delta \sin(t\Theta), \quad \text{for } t \in [0, 1], \tag{6}$$

*and the length of this geodesic corresponds exactly to the* projection distance *defined by*

$$d_{proj}(\mathcal{M}_k(\mathbf{X}), \mathcal{M}_k(\widetilde{\mathbf{X}})) = \left( \sum_{i=1}^{k} \theta_i^2 \right)^{1/2}. \tag{7}$$

Theorem 4 characterizes the geodesic path between two sets of temporal modes in time series data, establishing that the transformation between modes can be precisely expressed through a combination of trigonometric functions (proof in Appendix B).

This approach provides a quantitative and interpretable framework for assessing the performance of generative models in the context of time series data. We approximate our metric Equation 4 using the law of large numbers. We detail the computational steps in Algorithm 1.

---

**Algorithm 1:** Detailed Steps For Computing DMD-GEM.

---

**Inputs:** Number of samples $B$
**Initialize:** $m = 0$ ;
**foreach** $l = 1, \ldots, B$ **do**
    Sample a batch of original times series $\mathcal{X}$ and a batch of generated time series $\widetilde{\mathcal{X}}$.
    **foreach** $\mathbf{X}_i$ *in original data batch* $\mathcal{X}$ **do**
        **foreach** $\widetilde{\mathbf{X}}_j$ *in generated data batch* $\widetilde{\mathcal{X}}$ **do**
            1. Extract temporal modes $\mathcal{M}_k(\mathbf{X_i})$ of $\mathbf{X}_i$.
            2. Extract temporal modes $\mathcal{M}_k(\widetilde{\mathbf{X}_j})$ of $\widetilde{\mathbf{X}}_j$.
            3. Obtain orthonormal bases $\mathbf{Q}_i$ and $\widetilde{\mathbf{Q}}_j$ from
$$\mathcal{M}_k(\mathbf{X_i}) = \mathbf{Q_i R}_i, \mathcal{M}_k(\widetilde{\mathbf{X}_j}) = \widetilde{\mathbf{Q}_j}\widetilde{\mathbf{R}_j}.$$
            4. Obtain $\cos \boldsymbol{\Theta} = diag(\cos\theta_1, ..., \cos\theta_r)$ from
$$\mathbf{Q_i}^\top \widetilde{\mathbf{Q}_j} = \mathbf{U_{ang}}(\cos\boldsymbol{\Theta})\mathbf{V_{ang}}^T.$$
            5. Compute the dissimilarity matrix
$$\mathbf{C}_{ij} = d_P\left(\mathcal{M}_k(\mathbf{X_i}), \mathcal{M}_k(\widetilde{\mathbf{X}_j})\right) = \left(r - \textstyle\sum_{k=1}^{r}\cos^2\theta_k\right)^{1/2}.$$
        **end foreach**
    **end foreach**
    $D_{DMD-GEN} \leftarrow D_{DMD-GEN} + \min_{\gamma \in \Pi} \langle \gamma, \mathbf{C} \rangle_F$
**end foreach**
Return $D_{DMD-GEN}/B$

---

The values of the optimal mapping matrix $\gamma^\star = \arg\min_{\gamma \in \Pi} \langle \gamma, \mathbf{C} \rangle_p$ in Equation 4 reflect the extent to which the modes of each training time series are preserved within the generated time series. We can therefore use a guiding sampling technique based on $\gamma^\star$ for a faster and more effective learning.

**Time and Complexity.** The proposed DMD-GEN metric leverages DMD computation, Optimal Transport, and geodesic distance computation using principle angles, all of which contribute to

the overall computational complexity. The time complexity of each component can be estimated separately: *(i) DMD Complexity.* The time complexity of DMD primarily depends on computing the Singular Value Decomposition. Since we reduce the dimensionality to focus on dominant modes, making this step more efficient in practice, the time complexity to compute each $\mathcal{M}_k(\mathbf{X_i})$ and $\mathcal{M}_k(\widetilde{\mathbf{X}}_\mathbf{j})$ is $\mathcal{O}(n \times k^2)$. *(ii) Geodesic Distance Computation.* Calculating the principal angles between the subspaces involves QR decompositions and a SVD operation on the product of orthonormal matrices. Therefore, the complexity of computing the geodesic distance is $\mathcal{O}(k^3)$. *(iii) Optimal Transport Complexity.* To find the optimal mapping between the modes of original and generated time series, the complexity of solving the optimal transport problem using Sinkhorn's algorithm is $\mathcal{O}(B^2)$ where $B$ is the number of samples (Sinkhorn, 1967). Therefore, the overall complexity of DMD-GEN is $\mathcal{O}(B \times n^2 \times k^2 + B \times k^3 + B^2)$. The significant advantage of DMD-GEN is that it does not require additional training, which makes the overall approach computationally feasible compared to metrics that involve model retraining.

## 4 EXPERIMENTS

### 4.1 DATASETS

We evaluate the diversity of generative models across one synthetic dataset and three real-world datasets. The detailed statistics of each dataset can be found in Appendix C.

**Sine waves.** We generated a synthetic dataset consisting of two sets of sine waves to represent a bimodal distributed data. The data were generated using the following formula:

$$y(t) = A \cdot \sin(2\pi f t + \phi), \tag{8}$$

where $A$ is the amplitude, $f$ is the frequency, $t$ is the time variable and $\phi$ is the phase angle of the sine wave. Each mode consists of 2000 samples with phases being randomly chosen between 0 and $2\pi$. For all the samples, the duration is 2 seconds and the sample rate is 12, making the length of each sequence be 24. $A = 0.5$ and $f = 1$ Hz for the first mode, and $A = 5$ and $f = 0.5$ Hz for the second mode.

**Stock price.** To test our framework on a complex multimodal dataset, we used Google stocks data from 2004 to 2019, which was used in (Yoon et al., 2019). The data consists of 6 features which are daily open, high, low, close, adjusted close, and volume. The time series were then cut into sequences with length 24, following the setup in the work done by (Yoon et al., 2019).

**Energy.** We conducted experiments on UCI's air quality dataset (Vito, 2016) consisting of hourly averaged responses from an array of 5 metal oxide chemical sensors embedded in an Air Quality Chemical Multisensor Device in an Italian city. Data was recorded from March 2004 to February 2005 and consists of 28 features. Unlike the previous datasets, this one has an unimodal distribution. The data is cut into several sequences of length 7.

**Electricity Transformer Temperature and humidity (ETTh).** The ETTh dataset focuses on temperature and humidity data from electricity transformers (Zhou et al., 2021). It includes 2 years of data at an hourly granularity, providing detailed temporal information about transformer conditions.

### 4.2 BASELINE METRICS

We compared our proposed metric DMD-GEN with well-established time series evaluation metrics. Specifically, this comparison includes three key metrics:

**Predictive Score.** (Yoon et al., 2019) The predictive score evaluates how well a generative model captures the temporal dynamics of the original data. It involves training a model on the generated data and assessing its performance on a real dataset. A lower predictive score indicates that the generated data contains patterns that are more representative of the temporal patterns found in the original data.

**Discriminative Score.** (Yoon et al., 2019) The discriminative score measures the similarity between real and generated time series data by training a binary classifier to distinguish between them.

**Contextual Frechet Inception Distance (context-FID).** (Jeha et al., 2022) Context-FID is an adaptation of the Frechet Inception Distance (FID), a metric used to assess the quality of images

**sines**

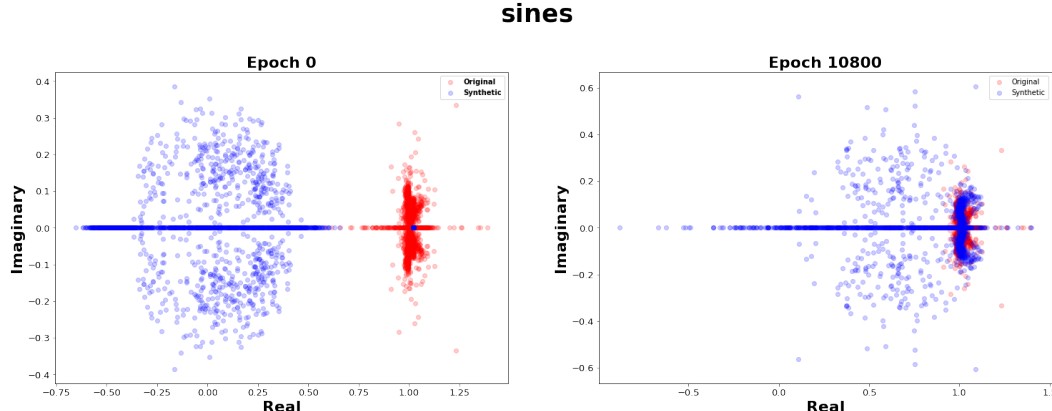

Figure 1: Comparison of DMD Eigenvalues between Original and Generated Time Series for DiffusionTS at Initial and Final Training Epochs on the dataset Sines.

**stock**

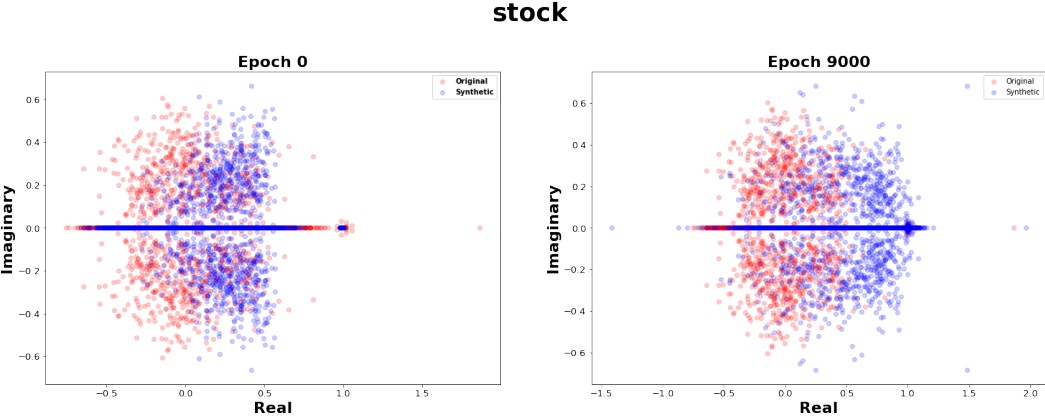

Figure 2: Comparison of DMD Eigenvalues between Original and Generated Time Series for DiffusionTS at Initial and Final Training Epochs on the dataset Stock.

created by a generative model (Heusel et al., 2017). For time series, context-FID measures the similarity between the real and generated data distributions by computing the Frechet distance between feature representations extracted from a time series feature encoder.

### 4.3 EVALUATION OF GENERATIVE MODELS USING DMD EIGENVALUES

Figures 1 and 2 present the DMD eigenvalues of the original training dataset and those generated by DiffusionTS at the initial and final stages of training for the time series Sines and Stock. At Epoch 0, the generated eigenvalues significantly deviate from the original ones, indicating that the generated time series lacks the dynamic properties of the original dataset. At the final training epoch, the DMD eigenvalues of the synthetic time series are much closer to the DMD eigenvalues of the original time series, highlighting that the model has successfully learned to capture the underlying temporal dynamics of the original dataset. This improvement demonstrates the capacity of DiffusionTS to learn and replicate complex temporal patterns through training. The DMD eigenvalues serve as effective indicators for identifying the similarity between the dynamics of the real and synthetic datasets, with the significant reduction in discrepancy pointing to improved quality and diversity of the generated data. The results of the same experiments conducted on other datasets, as well as the evolution of DMD eigenvalues throughout the training process of each of the generative models, can be found in Appendix E. DMD is highly useful for evaluating generative models for time series because it offers a clear and effective way to measure dynamic properties. It shows how well a model captures inherent dynamic patterns, offering a more insightful evaluation.

Table 1: Results on Multiple Time-Series Datasets. Highlighted results indicate the best performance. All four metrics agree on the best-performing model for each dataset. The symbol '-' denotes instances where the computation failed or crashed.

| Metric | Model | Sines | ETTh | Stock | Energy |
|---|---|---|---|---|---|
| Discriminative Score | TimeGAN | 0.03 (0.01) | **0.20 (0.03)** | **0.08 (0.04)** | **0.27 (0.04)** |
| | TimeVAE | 0.33 (0.02) | 0.50 (0.00) | 0.50 (0.00) | 0.50 (0.00) |
| | DiffusionTS | **0.02 (0.01)** | 0.50 (0.00) | 0.50 (0.00) | 0.50 (0.00) |
| Predictive Score | TimeGAN | 0.09 (0.00) | **12.39 (0.00)** | **6.40 (0.30)** | **24.01 (0.00)** |
| | TimeVAE | 0.12 (0.00) | 13.05 (0.03) | 27.12 (0.57) | 24.61 (0.06) |
| | DiffusionTS | **0.09 (0.00)** | 13.18 (0.01) | 17.78 (0.08) | 24.49 (0.07) |
| Context-FID | TimeGAN | 0.04 (0.01) | **0.40 (0.05)** | - | **11.92 (1.95)** |
| | TimeVAE | 5.01 (1.04) | 12.22 (1.15) | - | 135.27 (22.83) |
| | DiffusionTS | **0.01 (0.00)** | 11.65 (0.76) | - | 127.02 (13.68) |
| DMD-GEN | TimeGAN | 33.91 (1.75) | **20.96 (1.10)** | **0.73 (0.19)** | **44.57 (7.34)** |
| | TimeVAE | 31.65 (0.51) | 98.91 (0.68) | 4.02 (0.08) | 164.48 (0.44) |
| | DiffusionTS | **29.66 (0.34)** | 105.46 (0.82) | 13.62 (2.53) | 150.67 (0.97) |

## 4.4 Consistency of DMD-GEN with Established Metrics

Table 1 shows that DMD-GEN is consistent with other metrics, such as the Predictive Score, Discriminative Score, and Context-FID, when comparing generative models across different datasets. In all datasets, DMD-GEN's rankings align with those given by other metrics, effectively differentiating between generative models based on each metric values. A key advantage of DMD-GEN is that, unlike the other metrics, it doesn't require any training to evaluate the generated time series. This makes DMD-GEN a more efficient metric for practical use, as it reduces computational cost while maintaining consistent, reliable evaluation results.

## 4.5 Synthetically Analyzing Metrics Under Mode Collapse Settings

In order to study robustness under a range of mode collapse severities, we created a synthetic dataset where we can control the mode collapse severity. We generated a synthetic dataset consisting of $N = 1000$ time series, each sampled from one of two distinct generators, $\mathcal{G}_1$ and $\mathcal{G}_2$, defined as follows:

$$\mathcal{G}_1 = \left\{ (t, x) \mapsto \frac{a}{\cosh(x + b + 3)} \times \cos\left((c + 2.3) \cdot t\right) \mid x \in [-5, 5],\ t \in [0, 4\pi],\ (a, b, c) \sim \mathcal{U} \right\},$$

$$\mathcal{G}_2 = \left\{ (t, x) \mapsto \frac{2 + a}{\cosh(x)} \times \tanh(x) \times \sin\left((2.8 + b) \cdot t\right) \mid x \in [-5, 5],\ t \in [0, 4\pi],\ (a, b, c) \sim \mathcal{U} \right\},$$

where $\mathcal{U}$ denotes the uniform distribution over $[0, 1]$. Each time series is discretized to a length of $T = 129$ and a dimensionality of $d = 65$. Figure 3 in Appendix D illustrates examples of time series generated using $\mathcal{G}_1$ and $\mathcal{G}_2$. Time series generated by the same underlying generator are considered to belong to the same *mode*. To select the generator, we sample from a Bernoulli distribution with parameter $\lambda \in [0, 1]$. We choose generator $\mathcal{G}_1$ if $\lambda < \lambda_{\text{ref}}$, and generator $\mathcal{G}_2$ otherwise. The parameter $\lambda_{\text{ref}} = 0.5$ is the reference value where both modes are equally probable. At this reference value, there is no preference for either mode, avoiding mode collapse. We denote the resulting dataset by $\mathcal{D}_N(\lambda)$.

For each metric $m$, the value for the non-collapse scenario is represented by $m(\mathcal{D}_N(\lambda_{\text{ref}}), \mathcal{D}_N(\lambda_{\text{ref}}))$ and the value for the mode collapse scenario is represented by $m(\mathcal{D}_N(\lambda_{\text{ref}}), \mathcal{D}_N(\lambda))$ under different mode collapse severities $\mathcal{D}_N(\lambda)$, where $\lambda \neq \lambda_{\text{ref}}$. Since the metrics have different ranges, we compare metrics based on the performance defined as follows,

$$\text{Perf}(\lambda) = m(\mathcal{D}_N(\lambda_{\text{ref}}), \mathcal{D}_N(\lambda)) / m(\mathcal{D}_N(\lambda_{\text{ref}}), \mathcal{D}_N(\lambda_{\text{ref}})) - 1$$

We report the values of $\text{Perf}(\lambda)$ in Table 2, where we can see that the performance for the three benchmark metrics varies significantly in magnitude and sign as $\lambda$ changes.

Table 2: DMD-GEN demonstrates stable and robust performance for a wide range of mode collapse severities $\lambda$ compared to the benchmark metrics and it is more interpretable.

| Metric | $\lambda$ | | | | | |
|---|---|---|---|---|---|---|
| | 10% | 20% | 30% | 40% | 60% | 70% |
| Discriminative Score | +586.79 % | +443.40 % | +181.13 % | -20.75 % | -16.98 % | +143.40 % |
| Predictive Score | -0.54 % | -0.71 % | -0.83 % | -0.40 % | +0.35 % | +0.54 % |
| Context-FID | +36796.45 % | +18394.64 % | +8210.25 % | +1874.76 % | +1855.58 % | +7019.51 % |
| DMD-GEN | +681.03 % | +477.76 % | +312.22 % | +115.02 % | +114.92 % | +314.18 % |

Our proposed metric DMD-GEN is more stable and robust to the change in $\lambda$. Unlike Predictive and Discriminative scores, DMD-GEN is highly effective at detecting even small mode collapses. The performance of DMD-GEN, as well as Context-FID increases quickly when $\lambda$ deviates from the reference value $\lambda_{\mathrm{ref}}$. DMD-GEN demonstrates its ability to quickly detect minor discrepancies in generated modes. Despite not achieving the values of Context-FID, DMD-GEN is an easy-to-implement solution to identify mode collapse.

## 5 CONCLUSION

In this paper, we introduced a novel metric, DMD-GEN, specifically designed to evaluate generative models and quantify mode collapse in time series. Using Dynamic Mode Decomposition (DMD) and Optimal Transport, DMD-GEN provides a robust framework to evaluate the similarity of dynamic patterns between generated and original time series. Compared to existing metrics like Discriminative Score, and Predictive Score, DMD-GEN showed superior sensitivity in mode collapse detection. Furthermore, when comparing generative models, the results indicate that DMD-GEN consistently aligns with all other baseline metrics that require additional training, while DMD-GEN itself does not require any training, making it a more efficient alternative. DMD-GEN provides increased interpretability by decomposing the underlying dynamics into distinct modes, allowing for a clearer understanding of the preservation of essential time series characteristics. This highlights the potential of DMD-GEN as a crucial tool for advancing generative modeling techniques in time series analysis, promoting diversity in generated outputs. Future work can explore the integration of DMD-GEN into the training process, potentially using it as a guiding metric to improve model training dynamically.

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

# A   DYNAMICAL MODE DECOMPOSITION: DETAILS AND PROOFS

## A.1   THE LINK BETWEEN THE DMD OPERATORS IN CONTINUOUS AND DISCRETE CASES

Given a dynamical system $\dot{\mathbf{x}}(t) = \mathbf{f}(\mathbf{x}(t), t; \mu)$, we linearly approximate the dynamics using DMD using the operator $\mathcal{A} \in \mathbb{R}^{n \times n}$, i.e.

$$\forall t, \ \ \dot{\mathbf{x}}(t) = \mathcal{A}\mathbf{x}.$$

Discretizing time into intervals of $\Delta t$ and capturing snapshots accordingly, we establish the relationship between consecutive time steps in the following equation:

$$\forall k, \quad \mathbf{x}_{k+1} = \mathbf{x}_k + \mathcal{A}\mathbf{x}_k \Delta t = (I + \Delta t \mathcal{A}) \, \mathbf{x}_k. \tag{9}$$

For a time-step $\Delta t$ that is sufficiently small, we can employ the first-order Taylor expansion of the matrix $exp(\Delta t \mathcal{A})$, expressed as:

$$exp(\Delta t \mathcal{A}) \approx I + \Delta t \mathcal{A} \tag{10}$$

Therefore, from Equations 9 and 10, we conclude that:

$$\forall k, \quad \mathbf{x}_{k+1} \approx exp(\Delta t \mathcal{A})\mathbf{x}_k.$$

Thus,

$$\mathbf{A}^\star \approx exp(\Delta t \mathcal{A}).$$

## A.2   FEASIBLE SPECTRAL DECOMPOSITION OF THE DMD OPERATOR USING DIMENSIONALITY REDUCTION

Algorithm 2 presents the steps to compute the eigenvectors and eigenvalues of the DMD operator $\mathbf{A}^\star$ using Singular Value Decomposition (SVD) for dimensionality reduction.

## A.3   DMD EXPANSION

We will proof the closed formula $\forall k, \quad \mathbf{x_k} = \sum_{j=1}^{r} \phi_j \lambda_j^k b_j = \mathbf{\Phi} \mathbf{\Lambda^k} \mathbf{b}$, using recursion.

For $k = 0$, we have,

$$\begin{aligned}
\mathbf{x_0} &= \mathbf{I}\mathbf{x_0} \\
&= \mathbf{\Phi}\mathbf{\Phi}^\dagger \mathbf{x_0} \\
&= \mathbf{\Phi}\mathbf{b} \\
&= \mathbf{\Phi}\mathbf{\Lambda^0}\mathbf{b}
\end{aligned}$$

Let's now consider the equation hold for $k = 0, \ldots, m$, we have,

$$\begin{aligned}
\mathbf{x_{k+1}} &= \mathbf{A}^\star \mathbf{x_k} \\
&= \mathbf{A}^\star \mathbf{\Phi}\mathbf{\Lambda^k}\mathbf{b} \\
&= \mathbf{\Phi}\mathbf{\Lambda}\mathbf{\Lambda^k}\mathbf{b} \\
&= \mathbf{\Phi}\mathbf{\Lambda^{k+1}}\mathbf{b}.
\end{aligned}$$

Therefore, the equality holds for all $k \in \mathbb{N}$.

**Algorithm 2:** Dynamic Mode Decomposition

1. From collected snapshots of the system, build a pair of data matrices $(\mathbf{X}, \mathbf{X}')$.

$$\mathbf{X} = \left[ \begin{array}{cccc} | & | & & | \\ \mathbf{x}_0 & \mathbf{x}_1 & \cdots & \mathbf{x}_{m-1} \\ | & | & & | \end{array} \right], \mathbf{X}' = \left[ \begin{array}{cccc} | & | & & | \\ \mathbf{x}_1 & \mathbf{x}_2 & \cdots & \mathbf{x}_m \\ | & | & & | \end{array} \right]$$

The closed formula of optimal DMD operator is

$$\mathbf{A}^\star = \mathbf{X}' \mathbf{X}^\dagger$$

2. Compute the compact singular value decomposition (SVD) of $\mathbf{X}$:

$$\mathbf{X} \approx \mathbf{U\Sigma V}^\dagger$$

where $U \in \mathbb{C}^{n \times r}, \Sigma \in \mathbb{C}^{r \times r}, V \in \mathbb{C}^{m \times r}$ and $r \leq min(m,n)$ is the rank of $\mathbf{X}$. Therefore,

$$\mathbf{A}^\star = \mathbf{X}'\mathbf{V\Sigma}^{-1}\mathbf{U}^\dagger$$

3. Define a matrix

$$\tilde{\mathbf{A}} = \mathbf{U}^\dagger \mathbf{A}^\star \mathbf{U} = \mathbf{U}^\dagger \mathbf{X}'\mathbf{V\Sigma}^{-1},$$

since $U$ is a unitary matrix.

$\tilde{\mathbf{A}} \in \mathbb{R}^{r \times r}$ defines a low-dimensional linear model of the dynamical system on proper orthogonal decomposition (POD) coordinates.

4. Compute the eigen-decomposition of $\tilde{\mathbf{A}}$:

$$\tilde{\mathbf{A}}\mathbf{W} = \mathbf{W\Lambda},$$

where columns of $\mathbf{W} \in \mathbb{R}^{r \times r}$ are eigenvectors and $\mathbf{\Lambda} = diag(\lambda_1, \ldots, \lambda_r) \in \mathbb{R}^{r \times r}$ is a diagonal matrix containing the corresponding eigenvalues.

5. Return DMD modes $\mathbf{\Phi}$:

$$\mathbf{\Phi} = \mathbf{X}'\mathbf{V\Sigma}^{-1}\mathbf{W}.$$

Each column of $\mathbf{\Phi}$ is an eigenvector of $\mathbf{A}$ meaning a DMD mode $\phi_k$ corresponding to eigenvalue $\lambda_k$

# B  MATHEMATICAL PROOFS

## B.1  PROOF OF THEOREM 4 — DMD MODE GEODESIC

**Theorem 4**[DMD Mode Geodesic] Let $\mathcal{M}_k(\mathbf{X}), \mathcal{M}_k(\widetilde{\mathbf{X}}) \in \mathbb{R}^{n \times k}$ be matrices whose columns form orthonormal bases of two $k$-dimensional subspaces of $\mathbb{R}^n$. Let $\Theta = \mathrm{diag}(\theta_1, \theta_2, \ldots, \theta_k)$ be the diagonal matrix of principal angles between the subspaces spanned by $\mathcal{M}_k(\mathbf{X})$ and $\mathcal{M}_k(\widetilde{\mathbf{X}})$. Let $\Delta \in \mathbb{R}^{n \times k}$ be an orthonormal matrix such that

$$\mathcal{M}_k(\widetilde{\mathbf{X}}) = \mathcal{M}_k(\mathbf{X}) \cos(\Theta) + \Delta \sin(\Theta). \tag{11}$$

Then, the geodesic linking $\mathcal{M}_k(\mathbf{X})$ and $\mathcal{M}_k(\widetilde{\mathbf{X}})$ on the Grassmann manifold $\mathrm{Gr}(k, n)$ is given by

$$\gamma(t) = \mathcal{M}_k(\mathbf{X}) \cos(t\Theta) + \Delta \sin(t\Theta), \quad \text{for } t \in [0, 1], \tag{12}$$

and the length of this geodesic corresponds exactly to the *projection distance* defined by

$$d_{\mathrm{proj}}(\mathcal{M}_k(\mathbf{X}), \mathcal{M}_k(\widetilde{\mathbf{X}})) = \left( \sum_{i=1}^{k} \theta_i^2 \right)^{1/2}. \tag{13}$$

*Proof.* **Preliminaries and Definitions**

1. **Grassmann Manifold** $\mathrm{Gr}(k, n)$: The set of all $k$-dimensional linear subspaces of $\mathbb{R}^n$.

2. **Orthonormal Bases**: For a $k$-dimensional subspace $\mathcal{S} \subset \mathbb{R}^n$, an orthonormal basis is represented by an $n \times k$ matrix $Q$ with columns satisfying $Q^\top Q = I_k$, where $I_k$ is the $k \times k$ identity matrix.

3. **Principal Angles and Vectors**: Given two subspaces $\mathcal{S}_1$ and $\mathcal{S}_2$ with orthonormal bases $Q_1$ and $Q_2$, the principal angles $0 \le \theta_1 \le \theta_2 \le \cdots \le \theta_k \le \frac{\pi}{2}$ between them are defined recursively by

$$\cos(\theta_i) = \max_{\substack{\mathbf{u} \in \mathcal{S}_1 \\ \|\mathbf{u}\|=1}} \max_{\substack{\mathbf{v} \in \mathcal{S}_2 \\ \|\mathbf{v}\|=1}} \mathbf{u}^\top \mathbf{v}, \quad \text{subject to } \mathbf{u}^\top \mathbf{u}_j = 0, \ \mathbf{v}^\top \mathbf{v}_j = 0, \ j = 1, \ldots, i-1. \tag{14}$$

4. **Projection Distance**: The projection distance between $\mathcal{S}_1$ and $\mathcal{S}_2$ is defined as

$$d_{\mathrm{proj}}(\mathcal{S}_1, \mathcal{S}_2) = \left( \sum_{i=1}^{k} \theta_i^2 \right)^{1/2}. \tag{15}$$

**1. Computation of the Principal Angles**

Let $Q_1 = \mathcal{M}_k(\mathbf{X})$ and $Q_2 = \mathcal{M}_k(\widetilde{\mathbf{X}})$. Both $Q_1$ and $Q_2$ are $n \times k$ matrices with orthonormal columns.

We construct the matrix $C$ as follows:

$$C = Q_1^\top Q_2 \in \mathbb{R}^{k \times k}. \tag{16}$$

Since $Q_1^\top Q_1 = I_k$ and $Q_2^\top Q_2 = I_k$, $C$ captures the pairwise inner products between the basis vectors of $Q_1$ and $Q_2$.

We then perform the Singular Value Decomposition (SVD) of $C$:

$$C = U \Sigma V^\top, \tag{17}$$

where

- $U, V \in \mathbb{R}^{k \times k}$ are orthogonal matrices, i.e., $U^\top U = V^\top V = I_k$.

- $\Sigma = \mathrm{diag}(\sigma_1, \sigma_2, \ldots, \sigma_k)$ with $\sigma_i \ge 0$.

The singular values $\sigma_i$ of $C$ are the cosines of the principal angles between the subspaces:

$$\sigma_i = \cos(\theta_i), \quad \theta_i \in [0, \pi/2], \quad i = 1, \ldots, k. \tag{18}$$

This result stems from the fact that the SVD aligns the basis vectors of $U$ and $V$ to maximize the projections in the directions of the principal angles, which correspond to the largest cosines.

Since principal angles $\theta_i$ are defined in the range $[0, \pi/2]$, their cosines naturally lie in $[0, 1]$, matching the range of the singular values of $C$. Thus, the singular values encode the geometric relationship between the subspaces $U$ and $V$ in terms of the principal angles. This connection is fundamental to Grassmannian geometry, as it allows the distances and alignments between subspaces to be analyzed using the principal angles and their cosines.

**2. Construction of Orthonormal Bases Aligned with Principal Directions**

Define new orthonormal bases:

$$A = Q_1 U, \quad B = Q_2 V. \tag{19}$$

**Verification of Orthonormality:**

$$A^\top A = (Q_1 U)^\top (Q_1 U) = U^\top Q_1^\top Q_1 U = U^\top I_k U = U^\top U = I_k, \tag{20}$$

$$B^\top B = (Q_2 V)^\top (Q_2 V) = V^\top Q_2^\top Q_2 V = V^\top I_k V = V^\top V = I_k. \tag{21}$$

We then compute $A^\top B$:

$$A^\top B = (Q_1 U)^\top (Q_2 V) = U^\top Q_1^\top Q_2 V = U^\top C V = U^\top (U \Sigma V^\top) V \tag{22}$$

$$= U^\top U \Sigma V^\top V = I_k \Sigma I_k = \Sigma. \tag{23}$$

Thus, $A^\top B = \Sigma = \operatorname{diag}(\cos(\theta_1), \ldots, \cos(\theta_k))$.

**3. Decomposition of $B$ in Terms of $A$ and $\Delta$**

We aim to express $B$ as a linear combination of $A$ and another orthonormal matrix $\Delta$ that is orthogonal to $A$.

Let us define $\Delta$:

$$\Delta = (B - A \cos(\Theta)) \sin(\Theta)^{-1}, \tag{24}$$

where $\cos(\Theta) = \Sigma$ and $\sin(\Theta) = \operatorname{diag}(\sin(\theta_1), \ldots, \sin(\theta_k))$, and $\sin(\Theta)^{-1}$ denotes the diagonal matrix with entries $\sin(\theta_i)^{-1}$.

**Verification that $\Delta$ is Orthogonal to $A$:**

$$A^\top \Delta = A^\top (B - A \cos(\Theta)) \sin(\Theta)^{-1} \tag{25}$$

$$= (A^\top B - A^\top A \cos(\Theta)) \sin(\Theta)^{-1} \tag{26}$$

$$= (\Sigma - I_k \cos(\Theta)) \sin(\Theta)^{-1} \tag{27}$$

$$= (\cos(\Theta) - \cos(\Theta)) \sin(\Theta)^{-1} = 0. \tag{28}$$

**Verification that $\Delta$ is Orthonormal**:

First, we compute $\Delta^\top \Delta$:

$$\Delta^\top \Delta = \left((B - A \cos(\Theta)) \sin(\Theta)^{-1}\right)^\top \left((B - A \cos(\Theta)) \sin(\Theta)^{-1}\right) \tag{29}$$

$$= \sin(\Theta)^{-1} (B - A \cos(\Theta))^\top (B - A \cos(\Theta)) \sin(\Theta)^{-1}. \tag{30}$$

We compute the inner term:

$$(B - A \cos(\Theta))^\top (B - A \cos(\Theta)) = (B^\top - \cos(\Theta) A^\top)(B - A \cos(\Theta)) \tag{31}$$

$$= B^\top B - B^\top A \cos(\Theta) - \cos(\Theta) A^\top B \tag{32}$$

$$+ \cos(\Theta) A^\top A \cos(\Theta). \tag{33}$$

Since $A^\top A = I_k$, $B^\top B = I_k$, and $A^\top B = \Sigma = \cos(\Theta)$:

$$(B - A\cos(\Theta))^\top (B - A\cos(\Theta)) = I_k - \cos(\Theta)^\top \cos(\Theta) - \cos(\Theta)^\top \cos(\Theta) \quad (34)$$

$$+ \cos(\Theta)^\top \cos(\Theta)\cos(\Theta)^\top \cos(\Theta) \quad (35)$$

$$= I_k - \cos^2(\Theta) - \cos^2(\Theta) + \cos^4(\Theta) \quad (36)$$

$$= I_k - 2\cos^2(\Theta) + \cos^4(\Theta). \quad (37)$$

But since $\sin^2(\Theta) = I_k - \cos^2(\Theta)$, we can write:

$$I_k - 2\cos^2(\Theta) + \cos^4(\Theta) = (I_k - \cos^2(\Theta))^2 = \sin^4(\Theta). \quad (38)$$

Thus,

$$\Delta^\top \Delta = \sin(\Theta)^{-1} \sin^4(\Theta) \sin(\Theta)^{-1} = \sin^2(\Theta) I_k = I_k. \quad (39)$$

Therefore, $\Delta$ is orthonormal.

**Expressing $B$ in Terms of $A$ and $\Delta$:**

Using Equation equation 24, we have:

$$B = A\cos(\Theta) + \Delta\sin(\Theta). \quad (40)$$

## 4. Define the Geodesic Path

On the Grassmann manifold, the geodesic $\gamma(t)$ from $A$ to $B$ is given by:

$$\gamma(t) = A\cos(t\Theta) + \Delta\sin(t\Theta), \quad t \in [0, 1]. \quad (41)$$

**Verification of Endpoints**:

At $t = 0$:

$$\gamma(0) = A\cos(0 \cdot \Theta) + \Delta\sin(0 \cdot \Theta) = AI_k + \Delta \cdot 0 = A. \quad (42)$$

At $t = 1$:

$$\gamma(1) = A\cos(\Theta) + \Delta\sin(\Theta) = B. \quad (43)$$

Thus, $\gamma(t)$ is a continuous path on $\mathrm{Gr}(k, n)$ connecting $A$ and $B$.

**Relate Back to Original Bases**:

Recall that $A = Q_1 U = \mathcal{M}_k(\mathbf{X})U$ and $B = Q_2 V = \mathcal{M}_k(\widetilde{\mathbf{X}})V$.

Since $U$ and $V$ are orthogonal matrices, the subspaces spanned by $Q_1$ and $A$, and by $Q_2$ and $B$, are identical. Therefore, we can express the geodesic in terms of $\mathcal{M}_k(\mathbf{X})$ and $\Delta$.

**Expressing the Geodesic in Original Terms**:

Let us redefine $\Delta$ accordingly to absorb $U$ and $V$, so that we can write:

$$\gamma(t) = \mathcal{M}_k(\mathbf{X})\cos(t\Theta) + \Delta\sin(t\Theta). \quad (44)$$

## 5. Compute the Length of the Geodesic

The length $L$ of the geodesic $\gamma(t)$ is given by:

$$L = \int_0^1 \|\dot{\gamma}(t)\|_F \ dt, \quad (45)$$

where $\|\cdot\|_F$ denotes the Frobenius norm.

**Compute the Derivative $\dot{\gamma}(t)$:**

Since $\gamma(t) = \mathcal{M}_k(\mathbf{X})\cos(t\Theta) + \Delta\sin(t\Theta)$, we have:

$$\dot{\gamma}(t) = -\mathcal{M}_k(\mathbf{X})\Theta\sin(t\Theta) + \Delta\Theta\cos(t\Theta), \quad (46)$$

where we used the fact that the derivative of $\cos(t\Theta)$ with respect to $t$ is $-\Theta\sin(t\Theta)$, and similarly for $\sin(t\Theta)$.

**Compute the Squared Norm $\|\dot{\gamma}(t)\|_F^2$:**

$$\|\dot{\gamma}(t)\|_F^2 = \mathrm{Tr}\left(\dot{\gamma}(t)^\top \dot{\gamma}(t)\right) \tag{47}$$

$$= \mathrm{Tr}\left((-\mathcal{M}_k(\mathbf{X})\Theta\sin(t\Theta) + \Delta\Theta\cos(t\Theta))^\top (-\mathcal{M}_k(\mathbf{X})\Theta\sin(t\Theta) + \Delta\Theta\cos(t\Theta))\right) \tag{48}$$

$$= \mathrm{Tr}\left(\Theta^2\left(\sin^2(t\Theta)\mathcal{M}_k(\mathbf{X})^\top\mathcal{M}_k(\mathbf{X}) + \cos^2(t\Theta)\Delta^\top\Delta - \sin(t\Theta)\cos(t\Theta)\left(\mathcal{M}_k(\mathbf{X})^\top\Delta - \Delta^\top\mathcal{M}_k(\mathbf{X})\right)\right)\right). \tag{49}$$

Since $\mathcal{M}_k(\mathbf{X})^\top\mathcal{M}_k(\mathbf{X}) = I_k$, $\Delta^\top\Delta = I_k$, and $\mathcal{M}_k(\mathbf{X})^\top\Delta = 0$, the cross terms vanish, and we have:

$$\|\dot{\gamma}(t)\|_F^2 = \mathrm{Tr}\left(\Theta^2\left(\sin^2(t\Theta)I_k + \cos^2(t\Theta)I_k\right)\right) \tag{50}$$

$$= \mathrm{Tr}\left(\Theta^2 I_k\right) \tag{51}$$

$$= \sum_{i=1}^{k}\theta_i^2. \tag{52}$$

**Compute the Length $L$:**

Since $\|\dot{\gamma}(t)\|_F$ is constant with respect to $t$, we have:

$$L = \int_0^1 \|\dot{\gamma}(t)\|_F \, dt = \|\dot{\gamma}(t)\|_F \int_0^1 dt \tag{53}$$

$$= \left(\sum_{i=1}^{k}\theta_i^2\right)^{1/2} \cdot 1 \tag{54}$$

$$= \left(\sum_{i=1}^{k}\theta_i^2\right)^{1/2}. \tag{55}$$

**6. Length Equals the Projection Distance**

Comparing the computed length $L$ with the projection distance defined in Equation equation 15, we find:

$$L = d_{\mathrm{proj}}(\mathcal{M}_k(\mathbf{X}), \mathcal{M}_k(\widetilde{\mathbf{X}})) = \left(\sum_{i=1}^{k}\theta_i^2\right)^{1/2}. \tag{56}$$

On the Grassmann manifold, the geodesic distance between two subspaces is given by the length of the shortest path connecting them. This distance is intrinsically linked to the principal angles between the subspaces. The projection distance quantifies the separation between subspaces in terms of these principal angles.

By computing the squared norm of the derivative of the geodesic, we find that it equals the sum of the squares of the principal angles, which is the squared projection distance. Since the derivative's norm is constant, the total length of the geodesic over the interval $t \in [0, 1]$ is precisely the projection distance.

Therefore, the length of the geodesic $\gamma(t)$ connecting $\mathcal{M}_k(\mathbf{X})$ and $\mathcal{M}_k(\widetilde{\mathbf{X}})$ on the Grassmann manifold equals the projection distance between these two subspaces.

This completes the proof of Theorem 4.

$\square$

## C DATASETS AND IMPLEMENTATION DETAILS

### C.1 BASIC STATISTICS ON THE DATASETS

In Table 3, we present some basic statistics on the used datasets.

Table 3: Statistics of the four datasets used in our experiments.

| DATASET | SINE | STOCK | ENERGY | ETTH |
|---|---|---|---|---|
| #SAMPLES | 10,000 | 3,773 | 19,711 | 17,420 |
| DIMENSION | 5 | 6 | 28 | 8 |

### C.2 IMPLEMENTATION DETAILS

The experiments were conducted on an NVIDIA A100 GPU. We utilized the pyDMD package [1] in Python to compute the DMD eigenvalues and eigenvectors. For generating synthetic time series, we used the original settings and the official implementation of DiffusionTS[2], TimeGAN[3] and TimeVAE[4].

## D SYNTHETIC GENERATORS

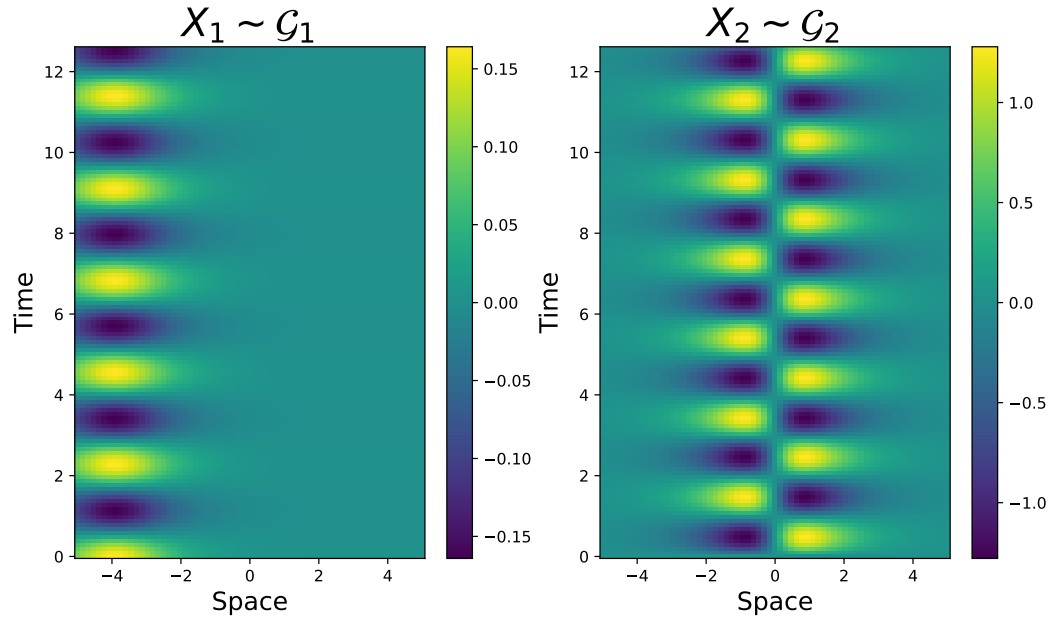

Figure 3: Examples of time series generated using the generators $\mathcal{G}_1$ and $\mathcal{G}_2$.

## E EVOLUTION OF THE DMD EIGENVALUES DURING TRAINING

In Figures 4, 5, 6, and 7, we plot the imaginary and real parts of the DMD eigenvalues of a 500 sample original and generated time series for each dataset.

---

[1]https://pydmd.github.io/PyDMD/

[2]https://github.com/Y-debug-sys/Diffusion-TS

[3]https://github.com/Y-debug-sys/Diffusion-TS

[4]https://github.com/zzw-zwzhang/TimeGAN-pytorch

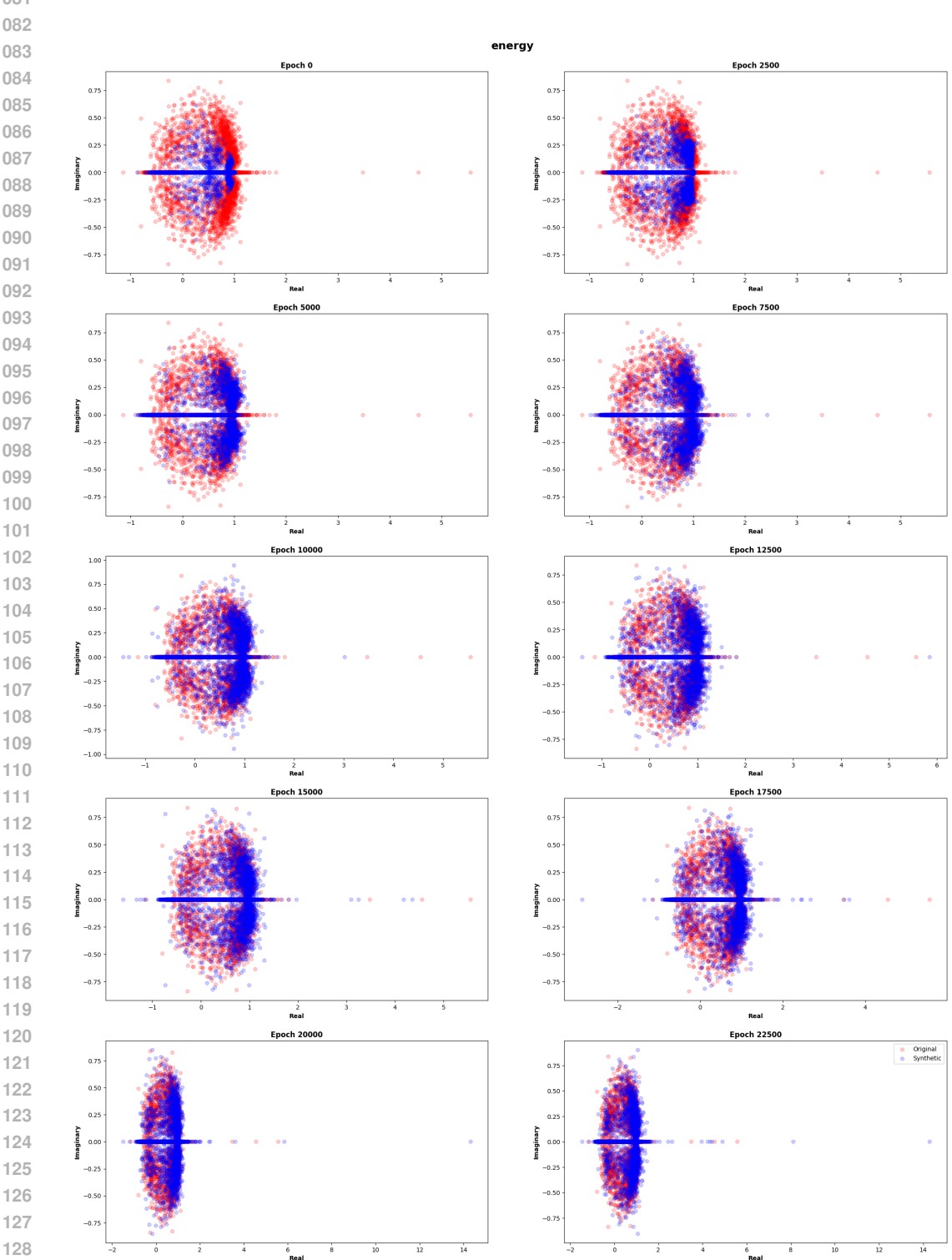

Figure 4: Comparison of DMD Eigenvalues between Original and Generated Time Series for DiffusionTS through Epochs on the dataset Energy.

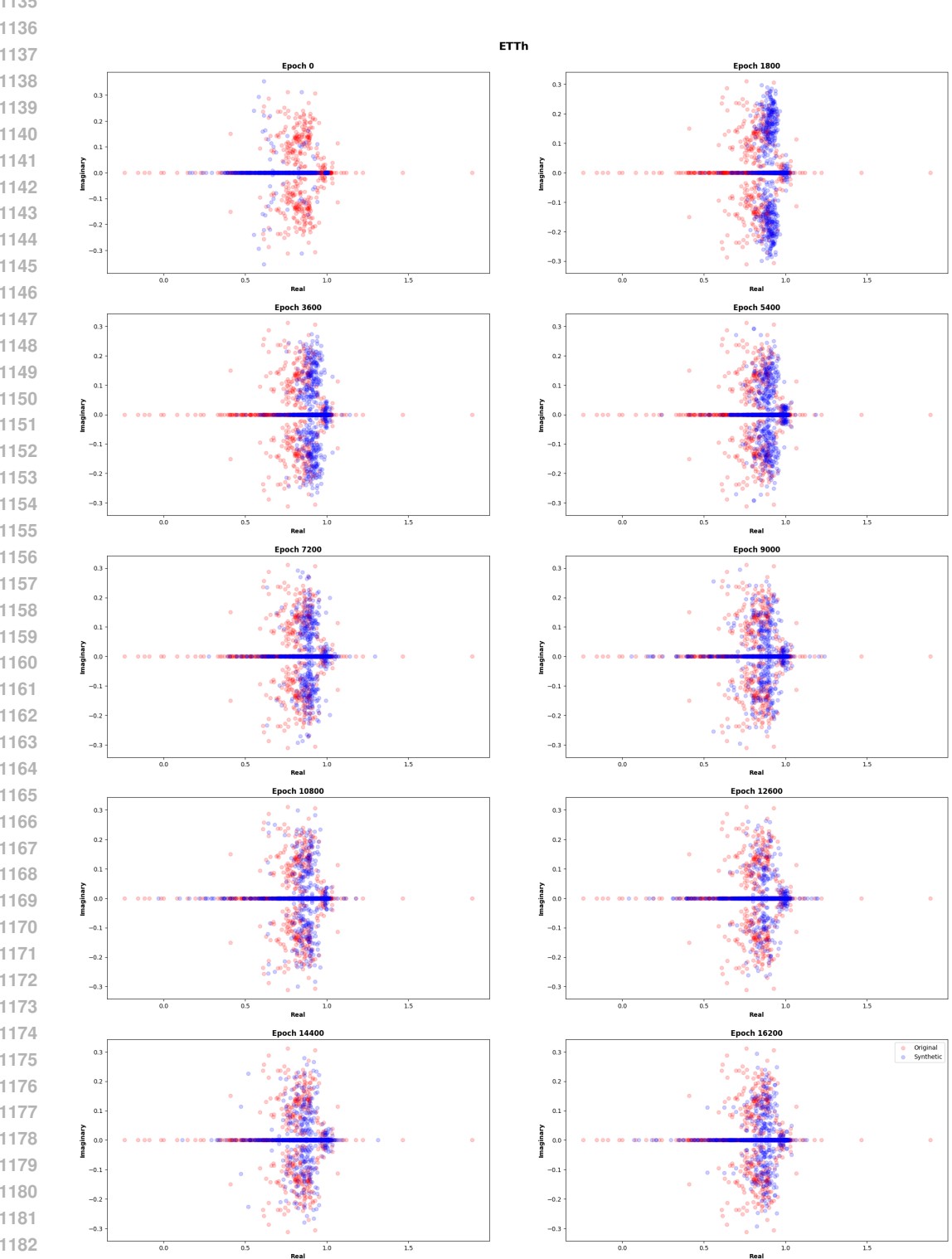

Figure 5: Comparison of DMD Eigenvalues between Original and Generated Time Series for DiffusionTS through Epochs on the dataset ETTh.

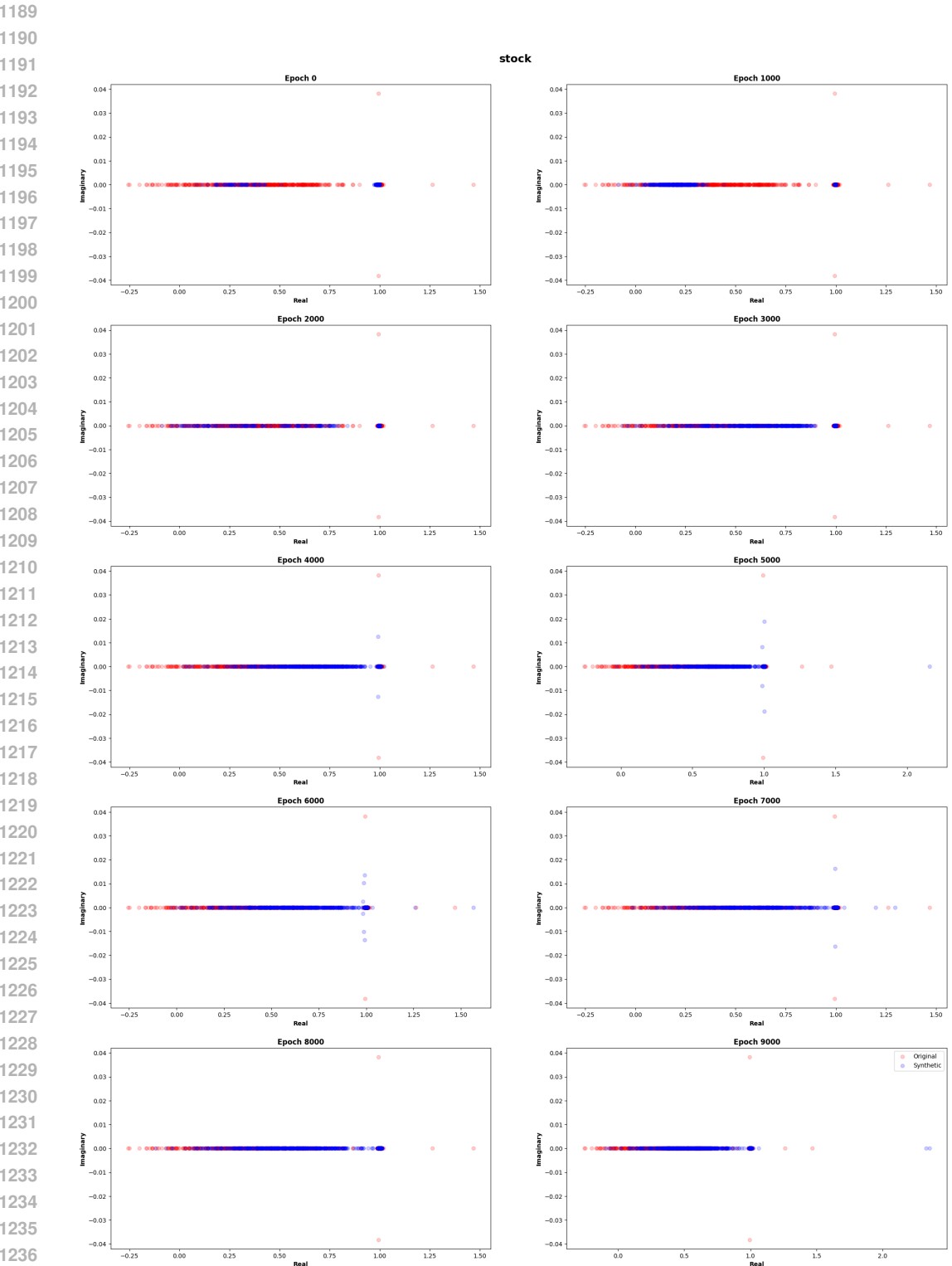

Figure 6: Comparison of DMD Eigenvalues between Original and Generated Time Series for DiffusionTS through Epochs on the dataset Stock.

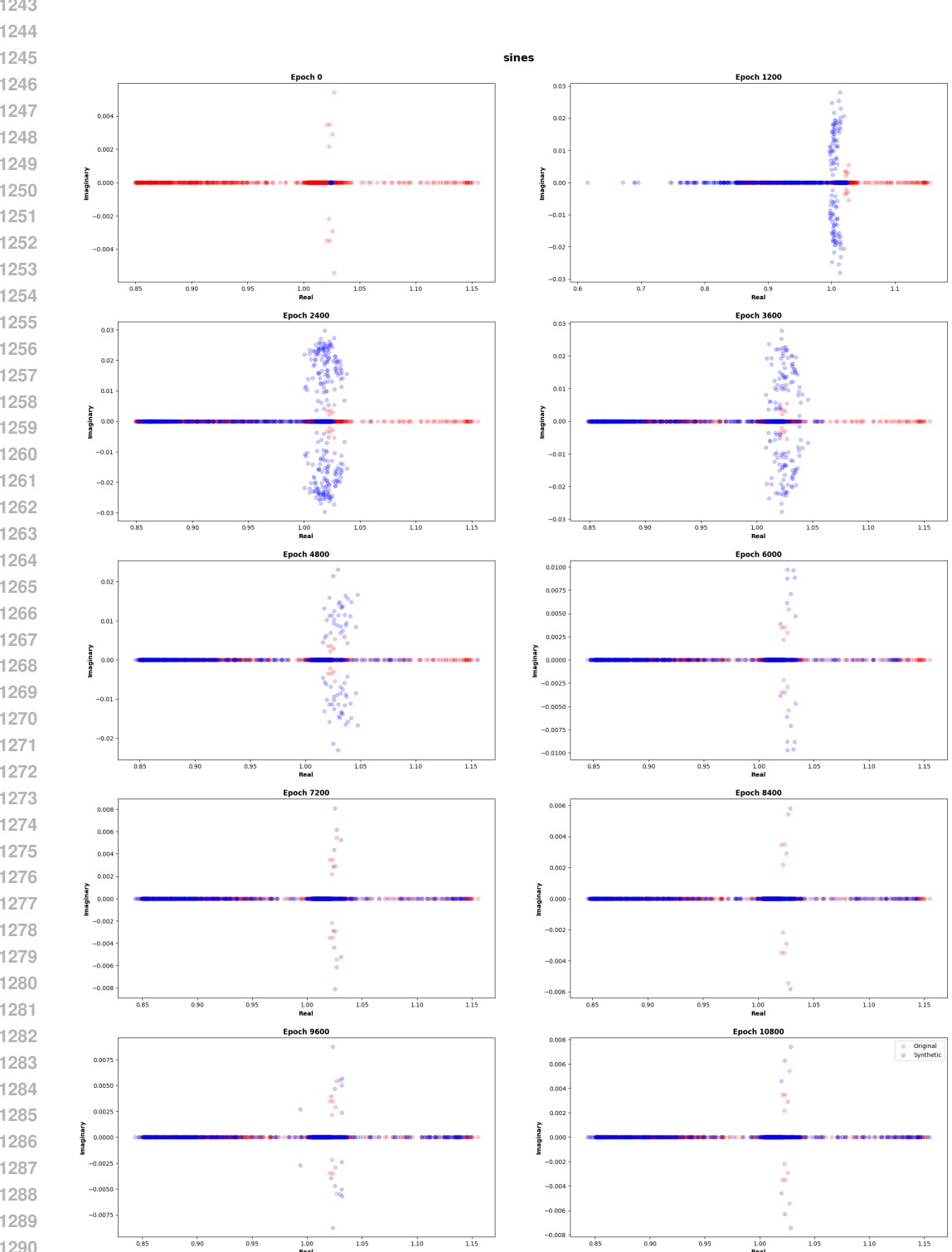

Figure 7: Comparison of DMD Eigenvalues between Original and Generated Time Series for DiffusionTS through Epochs on the dataset Sines.