# OpenReview forum: "Grassmannian Geometry Meets Dynamic Mode Decomposition in DMD-GEN: A New Metric for Mode Collapse in Time Series Generative Models"
_ICLR.cc/2025/Conference — Submitted to ICLR 2025_

### Official Review · Reviewer_cbWD · 2024-10-28

**Soundness:** 1
**Presentation:** 1
**Contribution:** 2
**Rating:** 3
**Confidence:** 3

**Summary:**

This work focuses on the time series generation problem. Specifically, they focus on the issue of "mode collapse" in time series generative model and propose a new notation of mode collapse and a new evaluation metric based on Dynamic Mode Decomposition (DMD) to quantify time series model collapse. They claim the proposed metric has connection/motivation/benefits from Grassmannian geometry and optimal transport. They provide some experiments on simple time series datasets to support their claims.

**Strengths:**

-

**Weaknesses:**

The main theme of this paper is confusing. The proposed metric seems to trailed for dynamical systems, with "highest frequency $\Delta t$" (`line 135`) known. Yet, the authors claim it to be a generic metric and test it on common time series datasets. Moreover, there are clear logic gaps in conveying the motivation/theoretical ground of the proposal. I find it hard to understand, and not very convincing. I detail reasons and clarification questions in below.

* Why Grassmannian?
* How does the propose method/metric handle the inherent noise in time series? Can you elaborate this from a data-driven perspective? (i.e., how the number of sample affect the noise robustness?)
* The experiments are not convincing. Please include more summary statistics of the generated data, and compare with baselines. For example, it's well-known that stock data have many important  summary statistics like volatility, moving average...e.t.c. It's only fair to show that the proposal achieve more similar summary statistics to ensure its efficacy.
* Is Thm. 3.4 your own result or from previous work? If it's from previous work, can you use lemma and specify source in lemma title instead?

**Questions:**

see above Weaknesses

---

> ### Author Response · Authors · 2024-11-24
> **Response to Reviewer cbWD (1/2)**
>
> Thank you very much for your review. In what follows, we address the raised questions and weaknesses point-by-point.
>
>
> ***Response to Weakness 1: On the Term ''Grassmannian''***
>
> We employed the Grassmannian manifold as the foundational mathematical tool for our proposed metric due to its robust capability to compare subspaces. Specifically, the Grassmannian framework allows us to construct a metric by measuring the distance between subspaces. In our approach, these subspaces are constructed using the eigenvectors of both the original and generated time series. This enables a comparison within the dynamical mode space, providing a quantitative assessment of whether the generated time series faithfully replicates the dynamical characteristics of the original. By leveraging the Grassmannian structure, our metric captures the geometric relationships of the dominant modes, ensuring a generic and invariant comparison mechanism applicable across diverse time series datasets.
>
> ***Response to Weaknesses 2: How does the proposed method handle noise in time series?***
>
> We appreciate the opportunity to elaborate on how our approach addresses noise from a data-driven perspective, particularly focusing on the influence of sample size on noise robustness. Below, we detail the mechanisms and strategies employed by DMD-GEN to ensure reliable and stable evaluations of generative models, even in the presence of noisy data.
>
>
>  1. ***Emphasis on Dominant Modes:*** DMD-GEN focuses on extracting and analyzing the dominant modes of the time series data. These dominant modes capture the primary underlying dynamics of the system, which are inherently less susceptible to random noise. By concentrating on these significant components, the method effectively reduces the influence of noise present in the data.
>  2. ***Batch Processing and Averaging:*** Each batch processed by DMD-GEN represents a snapshot of the time series dynamics. By aggregating information across multiple batches, the approach averages out random noise fluctuations. This averaging process enhances the stability and reliability of the evaluation by mitigating the impact of transient noise.
>  3. ***Impact of Sample Size on Noise Robustness:*** From a data-driven perspective, the number of samples plays a crucial role in noise robustness. As the sample size increases, the law of large numbers ensures that random noise effects diminish, while the true signal becomes more pronounced. Consequently, larger sample sizes lead to more accurate and noise-resistant metric estimates, enhancing the robustness of DMD-GEN against inherent noise in the time series.
>  4. ***Multiple Sampling Iterations:*** DMD-GEN computes the evaluation metric over numerous sampling iterations. This repeated sampling further stabilizes the results by ensuring that transient noise does not disproportionately affect the final metric. The cumulative effect of multiple iterations provides an unbiased estimate of the true metric value, reinforcing the method’s resilience to noise-induced fluctuations.
>
> Through a combination of focusing on dominant modes, batch processing with averaging, leveraging large sample sizes, and employing multiple sampling iterations, DMD-GEN ensures robust and stable evaluations of generative models on noisy time series data. These data-driven strategies collectively mitigate the impact of inherent noise, providing reliable metric estimates.
>
> We have updated the manuscript accordingly to incorporate these clarifications and additional explanations based on your feedback.

---

> > ### Author Response · Authors · 2024-11-24
> > **Response to Reviewer cbWD (2/2)**
> >
> > ***Response to Weakness 3: Adding Statistics to Evaluate the Generated Data***
> >
> > We would like to emphasize that while traditional summary statistics such as volatility and moving averages are widely used for analyzing financial time series, they primarily address univariate characteristics and do not capture the complexities inherent in multivariate time series data. For example:
> >
> >   - ***Limitations of Univariate Statistics:*** Statistics like volatility and moving averages focus on individual time series and do not account for interactions between variables (e.g., open price, close price, and trading volume). These interactions are critical in financial datasets, where interdependencies among variables play a significant role in overall dynamics.
> >
> >   - ***Suitability for Multivariate Generative Models:*** Evaluating multivariate time series requires metrics that can capture joint distributions, temporal correlations, and cross-variable relationships. Metrics tailored to these aspects are more appropriate for assessing the performance of generative models like DMD-GEN, particularly in addressing challenges such as mode collapse.
> >
> > We have addressed this aspect in greater detail in the revised manuscript.
> >
> >
> > ***Response to Weakness 4: Proof of Theorem 3.4***
> >
> > We appreciate your feedback regarding Theorem 3.4 and its attribution.
> >
> > Theorem 3.4 is derived from the theory of Grassmannian geometry, which provides a mathematical framework for analyzing subspaces using principal angles and geodesics. While the underlying concepts are established in the literature, our work extends these ideas in the context of defining the projection distance used in DMD-GEN.
> >
> > To ensure clarity and proper attribution, we have revised the manuscript as follows:
> >
> >  - Explicitly cited the foundational sources of Grassmannian geometry in the theorem’s title and body.
> >  - Clarified our contributions in applying these principles within the specific framework of our method.
> >  - Included the complete derivation in Appendix B to provide transparency and facilitate reproducibility.
> >
> > We believe these changes address your concern and improve the overall clarity of the manuscript. Thank you for your valuable suggestion.

---

> > > ### Comment · Reviewer_cbWD · 2024-11-24
> > >
> > > Thanks for your response.
> > >
> > > Is Thm 3.4 (Thm 4 in the latest revision) your original contribution or not?

---

> ### Author Response · Authors · 2024-11-24
> **Clarifying the Novelty and Role of Theorem 4 in Our Contribution**
>
> Thank you for your question.
>
> Yes, Theorem 4 is entirely our original contribution. Theorem 4 is fundamental to our proposed DMD-GEN metric, as it characterizes the geodesic path between the subspaces spanned by the temporal modes of the original and generated time series on the Grassmannian manifold. We have provided a detailed proof in Appendix B to substantiate this contribution.
>
> By characterizing the geodesic path, Theorem 4 naturally induces a metric on the space of dynamical modes. We use this metric to compute the optimal transport cost between temporal modes, which is the central idea behind Algorithm 1.
>
> Algorithm 1, based on this metric, allows us to explicitly compute and quantify mode collapse, particularly when a generated mode (in the generated time series) cannot be optimally transported to match an original mode (in the original time series). We believe this theoretical development is novel and provides a valuable framework for analyzing mode collapse in time series generation.

---

> ### Comment · Reviewer_cbWD · 2024-11-25
>
> Thank you for the clarification, detailed responses, and revision efforts.
>
> However, I believe the draft still requires further polishing and is not yet ready for publication.
>
> The primary issue is a lack of clarity. This prevents readers from understanding and assessing the work's contributions.
>
> Below are my suggestions for improvement:
>
> * Math Writing: As a math/theory-heavy paper, adhering to proper mathematical writing standards is essential for clarity and rigor. I recommend reviewing the mathematical writing guide https://jmlr.csail.mit.edu/reviewing-papers/knuth_mathematical_writing.pdf to refine the presentation of definitions, theorems, proofs, and equations.
>
> * Overall Writing: The academic professionalism of the draft needs improvement to better convey its originality, contribution, and correctness. For instance, the paper would benefit from clear and concise descriptions of the key results, along with improved structure and flow between sections. While I attempted to compile specific examples of writing issues, there are too many to enumerate comprehensively. A few thorough proofreading pass, potentially involving external feedback or professional editing, could help address these concerns.
>
> I hope these suggestions are helpful in guiding future revisions. Thank you again for your hard work and effort in improving the manuscript.

---

### Official Review · Reviewer_avZD · 2024-11-01

**Soundness:** 2
**Presentation:** 2
**Contribution:** 2
**Rating:** 5
**Confidence:** 3

**Summary:**

The paper proposes DMD-GEN, a method for evaluating mode collapse in timeseries generative models.
The method is based on evaluating Grassmanian geometry for dynamic mode decomposition (DMD) for timeseries.
In experiments, DMD-GEN is compared to baselines on synthetic and real timeseries.

**Strengths:**

1. Novelty. To my knowledge, this is the first paper which studies directly a problem of mode collapse in timeseries.
2. A diverse set of timeseries generative models is studied (TimGAN, DiffusionTS, TimeVAE).

**Weaknesses:**

1. line 136: "Assuming uniform sampling in time, we approximate the dynamical system linearly as $x_{k+1} \approx A x_k$."
It seems to be an unrealistic assumption for real data.
2. Limited novelty. DMD was proposed before.
3. I don't understand, why DMD-GEN in superior to alternatives in section 4.5. For example, ContextFID also changes monotonically.
4. A comparison with an MTopDiv (Barannikov et al. 2021) is missing. MTopDiv can also be applied to evaluation of timeseries generative models.
5. You claim "This approach not only quantifies the preservation of essential dynamic characteristics but also provides interpretability by pinpointing which modes have collapsed." I can't find a proof that your method can pinpoint "which modes have collapsed.
6. You claim that "This work offers for the first time a definition of mode collapse for time series..."
The definition of mode collapse is quite basic - it means that a particular mode is present in training data, but absent in generated data.

Barannikov, S., Trofimov, I., Sotnikov, G., Trimbach, E., Korotin, A., Filippov, A., & Burnaev, E. (2021). Manifold Topology Divergence: a Framework for Comparing Data Manifolds. Advances in neural information processing systems, 34, 7294-7305.

**Questions:**

1.  "The results demonstrate that DMD-GEN correlates well with traditional evaluation metrics for static data while offering the advantage of applicability to dynamic data". What is the difference between dynamic and static data?
2. What does "||" mean in line 146, line 203?
3. Do you study univariate or multivariate timeseries?
4. What is the purpose of "x" in generators in section 4.5?

**Conclusion**.
Overall, I think that the paper raises an important research question: how to evaluate mode collapse in timeseries generative models.
But experiments providing the superiority w.r.t. baselines are not convincing and there are only few of them.
Some claims in abstract are not supported with strong evidence.
Overall, my opinion is that the paper contains some interesting ideas (application of DMD to evaluation of timeseries generative models),
but it must be accompanied with a stronger empirical evidences, including not only synthetic mode collapse (see examples of such experiments in Barannikov et al., 2021).


**Post rebuttal**. I am thankful to authors for the detailed responses and conducting additional experiments. Sorry for the delay in my answers.
I think that the discussion helped to improve that manuscript. I am updating my rating. The final rating is defined by the overall novelty and impact of your method.

---

> ### Author Response · Authors · 2024-11-24
> **Response to Reviewer avZD (1/3)**
>
> *We thank reviewer avZD for their thoughtful comments and insightful questions. Below, we address the raised questions and weaknesses point-by-point.*
>
>
> ***Response to Weakness 1: Unrealistic assumption of approximating the dynamical system linearly***
>
> We appreciate the reviewer's concern regarding the linear approximation of dynamical systems. The Koopman operator theory offers a rigorous mathematical framework for representing nonlinear dynamical systems as infinite-dimensional linear operators acting on a space of observables. This perspective enables the use of linear analysis techniques to study nonlinear dynamics.
>
> DMD leverages this theory by providing a data-driven approximation of the Koopman operator within a finite-dimensional subspace. By assuming uniform sampling in time, we facilitate this approximation, allowing DMD to decompose complex nonlinear behaviors into linear combinations of modes. This approach has been widely adopted to simplify and interpret complex nonlinear systems effectively.
>
> Therefore, the linear approximation inherent in DMD is a well-established method that remains valid and powerful for capturing the essential dynamics of real-world data, despite inherent nonlinearities.
>
> We have revised line 136 to better explain and further clarify this point.
>
>
>
>
> ***Response to Weakness 2: Limited Novelty (DMD was proposed before)***
>
> While DMD is a well-established technique, our work introduces a new theoretical framework that significantly extends its application:
>
>  - ***New Distance Measurement between DMD Modes:*** We propose a method to measure distances between DMD modes using Grassmannian geometry and Optimal Transport theory, which has not been explored before.
>  - ***Robust Comparison of Subspaces:*** By leveraging the subspaces spanned by the dominant DMD eigenvectors of real and generated time series, we enable robust and interpretable comparisons between them.
>  - ***Principal Angles as a Metric:*** We define principal angles between subspaces and use them as a metric, providing a mathematically rigorous and scalable approach to evaluate generative models on time series data.
>
>
> Our contribution lies in the integration of DMD with advanced mathematical tools to create a new evaluation methodology for generative time series models. This represents a significant advancement beyond the existing use of DMD.
>
>
>
> ***Response to Weakness 3: Superiority of DMD-GEN Compared to ContextFID***
>
> We acknowledge that both DMD-GEN and ContextFID exhibit monotonic changes. However, DMD-GEN offers several key advantages over ContextFID:
>
>  - ***Computational Efficiency:*** DMD-GEN requires no training, making it computationally efficient and scalable to large datasets. In contrast, ContextFID involves training procedures that can be resource-intensive.
>  - ***Interpretability of Temporal Dynamics:*** DMD-GEN explicitly captures temporal coherence by decomposing time series data into DMD modes. This allows us to identify which specific temporal modes are preserved or collapsed by the generative model, providing insights not available with ContextFID or with other metrics as well.
>  - ***Robustness in Capturing Temporal Coherence:*** While ContextFID measures static similarity, it does not account for temporal dynamics. DMD-GEN focuses on the temporal aspects of the data, offering a more nuanced evaluation of generative models that produce sequential or time-dependent outputs.
>
> By leveraging these advantages, DMD-GEN provides a more comprehensive and interpretable assessment of generative models, particularly in capturing and analyzing temporal dynamics, which is beyond the scope of ContextFID.

---

> > ### Author Response · Authors · 2024-11-24
> > **Response to Reviewer avZD (2/3)**
> >
> > ***Response to Weakness 4: Comparison with MTopDiv***
> >
> > We appreciate the reviewer's suggestion to include MTopDiv (Barannikov et al. 2021) as a baseline for comparison. While MTopDiv is an effective metric for comparing data manifolds, it is not ideally suited for multivariate time series data that encompass both temporal and spatial dimensions. Applying MTopDiv to time series requires flattening the data into a collection of independent samples, which disregards the essential temporal structure and the evolving interactions between features over time.
> >
> > Nonetheless, for completeness, we conducted experiments using MTopDiv on the Energy, ETTh, and Sines datasets to compare the performance of TimeVAE, TimeGAN, and DiffusionTS. Accordingly, we have updated the experimental section in our paper to include MTopDiv as an additional baseline. The MTopDiv results are summarized in the table below.
> >
> > | Dataset | TimeVAE        | TimeGAN        | DiffusionTS    |
> > |---------|----------------|----------------|----------------|
> > | Energy  | 424.34 (19.75) | 467.35 (49.10) | 402.42 (34.53) |
> > | ETTh    | 116.23 (4.96)  | 130.85 (8.80)  | 116.51 (5.52)  |
> > | Sines   | 7.72 (0.18)    | 4.92 (0.32)    | 4.53 (0.13)    |
> >
> >
> > The results indicate substantial variability in the standard deviations for each model across datasets, which hampers the ability to make meaningful comparisons. This high variability suggests that MTopDiv may not be a reliable metric for evaluating multivariate time series generative models, as it fails to capture the temporal dependencies intrinsic to the data.
> >
> >
> >
> >
> > ***Response to Weakness 5: Identification of Collapsed Modes***
> >
> > Thank you for highlighting the need for clarification on how our method pinpoints collapsed modes. Our approach, DMD-GEN, utilizes the optimal transport (OT) plan $\gamma^*$ to establish a mapping between the dominant modes of the real and generated time series data.
> >
> > The OT plan $\gamma^*$ is obtained by minimizing the transportation cost between the distributions of modes in the original and generated batches. Specifically, for each mode $\phi_i$ in the original batch, we seek the best matching mode $\psi_j$ in the generated batch that minimizes the cost function in the OT framework. This process is formalized as:
> >
> > \begin{equation*}
> > \begin{aligned}
> > \gamma^* = \underset{\gamma}{\arg\min} \quad & \sum_{i,j} c(\phi_i, \psi_j) \, \gamma_{ij} \\
> > \text{subject to} \quad & \sum_j \gamma_{ij} = p_i \quad \text{for all } i \\
> > & \sum_i \gamma_{ij} = q_j \quad \text{for all } j \\
> > & \gamma_{ij} \geq 0
> > \end{aligned}
> > \end{equation*}
> >
> > Here, $c(\phi_i, \psi_j) $ represents the cost of transporting mode $ \phi_i $ to mode $ \psi_j $, measuring the dissimilarity between the modes. The variable $\gamma_{ij} $ denotes the transport plan between modes $ \phi_i $ (from the original data) and $ \psi_j $ (from the generated data). The terms $ p_i $ and $ q_j $ are the probability distributions over the modes in the original and generated batches, respectively.
> >
> > Modes in the original data that either have a high transportation cost or lack a corresponding match (i.e., $\gamma_{ij} = 0$ for all $j$) are identified as collapsed modes. This identification is possible because such modes are either poorly represented or entirely missing in the generated data.
> >
> > By analyzing the OT plan $\gamma^*$, DMD-GEN not only quantifies the fidelity of generated modes but also provides interpretability by explicitly pinpointing which specific modes have collapsed. We have expanded on this methodology and included illustrative examples in Section 3.3 of the revised manuscript to enhance clarity.
> >
> > ***Response to Weakness 6: Claim of Novel Definition for Mode Collapse***
> >
> > We appreciate the reviewer's observation regarding the basic definition of mode collapse. Our contribution lies in the novel adaptation of this concept specifically to time series data. In domains such as image generation, modes correspond to discrete and easily identifiable categories (e.g., ''dog'' or ''cat''), facilitating straightforward detection of mode collapse. In contrast, time series data lacks such discrete classes; modes represent complex dynamic patterns, such as trends, oscillations, and seasonal behaviors, that are inherently more challenging to formalize and isolate.
> >
> > To address this challenge, our work employs DMD to decompose time series into their dominant spatio-temporal patterns. This methodology provides a robust and interpretable framework for defining and quantifying modes within time series data. By doing so, we enable the identification and measurement of mode collapse in a domain where traditional class-based definitions are inapplicable.
> >
> > To the best of our knowledge, this is the first work to offer a formal definition of mode collapse tailored specifically for time series data. This advancement addresses a significant gap in the literature and enhances the understanding of generative modeling in the context of time series.

---

> > > ### Author Response · Authors · 2024-11-24
> > > **Response to Reviewer avZD (3/3)**
> > >
> > > ***Response to Question 1: What is the difference between dynamic and static data?***
> > >
> > > Static Data: Independent samples without temporal or sequential dependencies—for example, individual images in datasets like MNIST or CIFAR-10.
> > >
> > > Dynamic Data: Samples with temporal or sequential relationships, such as time series or video sequences, where the order and transitions between data points are significant.
> > >
> > > Our work highlights that while traditional evaluation metrics are suitable for static data, they may not adequately assess models generating dynamic data. DMD-GEN addresses this by capturing temporal dynamics, enabling comprehensive evaluation of generative models on dynamic datasets. It correlates well with traditional metrics on static data and extends their applicability to dynamic data.
> > >
> > >
> > >
> > > ***Response to Question 2: Meaning of ``$\left| \right|$'' in Lines 146 and 203***
> > >
> > > The symbol ``$\left| \right|$'' denotes the concatenation operation for vectors or matrices in our manuscript. Specifically, in Line 146, it represents the concatenation of eigenvectors to form the temporal modes matrix $\mathcal{M}_k(X)$. Similarly, in Line 203, it is used to denote concatenation when discussing the Grassmannian manifold.
> > >
> > > To enhance clarity, we have added an explicit explanation of this notation in Section 2.2 of the revised manuscript.
> > >
> > >
> > >
> > > ***Response to Question 3: Univariate vs. Multivariate Time Series***
> > >
> > > We study multivariate time series in our work. Multivariate time series analysis is crucial for capturing the complex interactions and dependencies among multiple variables over time, providing a more realistic representation of dynamic systems compared to univariate time series analysis. Specifically, DMD is particularly powerful when applied to multivariate data, as it decomposes the system into dominant modes that span the entire feature space, thereby capturing the essential dynamics across all variables. We will clarify our focus on multivariate time series in the revised manuscript.
> > >
> > > ***Response to Question 4: Purpose of ''x'' in Generators in Section 4.5***
> > >
> > > We appreciate the reviewer's question regarding the role of ''x'' in our synthetic data generators.
> > >
> > > In Section 4.5, the variable ''x'' represents a spatial coordinate in the generators $\mathcal{G}_1$ and $\mathcal{G}_2$. Its inclusion is essential for introducing spatial variability into the generated data. Specifically, "x" serves as a parameter that enhances the dimensionality of the synthetic time series, enabling the modeling of complex spatiotemporal patterns. This allows the generated time series to more accurately mimic realistic multivariate time series, where multiple features evolve over both time and space.
> > >
> > >
> > > *Given the brevity of your review, we hope that our additional explanations have helped you understand our work more deeply.*

---

### Official Review · Reviewer_ue8P · 2024-11-04

**Soundness:** 2
**Presentation:** 3
**Contribution:** 2
**Rating:** 6
**Confidence:** 2

**Summary:**

This paper presents DMD-GEN, a novel metric for assessing mode collapse in generative models for time series data, specifically using Dynamic Mode Decomposition (DMD). The study addresses a gap in mode collapse research, which has predominantly focused on image data, by proposing a time-series-specific approach. DMD-GEN leverages Optimal Transport to measure dynamic similarities between original and generated data, providing an interpretable metric to quantify the extent of mode collapse. The authors validate DMD-GEN on synthetic and real-world datasets using various generative models, demonstrating its consistency with traditional metrics offering computational efficiency without additional training.

**Strengths:**

The introduction of DMD-GEN as a time-series-specific metric for mode collapse is a **novel** contribution. This approach brings a new perspective to evaluating time series generative models by capturing coherent dynamic patterns, which is less explored.

The methodology is robust and **comprehensive**, integrating techniques from DMD and Optimal Transport. The authors provide a clear mathematical foundation for their approach, along with thorough experimental validation across diverse datasets. The authors also provide the **public code** of the DMD-GEN implementation for practical use.

DMD-GEN offers both **computational efficiency** and interpretability, making it a potentially valuable tool in time-series modeling. Its ability to detect and quantify mode collapse without retraining the model makes it practical for real-time applications.

**Weaknesses:**

While the experiments conducted demonstrate the effectiveness of DMD-GEN, the paper could benefit from a **broader range of datasets and generative models** to strengthen its generalizability. More extensive empirical validation could enhance the paper's claims about the metric's superiority.

While DMD-GEN is computationally efficient compared to metrics that require retraining, calculating DMD-GEN could still be resource-intensive on **large datasets** or **high-dimensional** time series. Discussing the limitations of DMD-GEN in terms of scalability would be valuable.

**Questions:**

Would the authors consider evaluating DMD-GEN on larger datasets and more complex and longer time-series data? This would strengthen the paper's claims regarding the metric's applicability.

Can the authors discuss potential scalability issues when applying DMD-GEN to larger or more complex datasets? For example, it could be useful to measure memory footprint and computational times as dataset size changes, while employing a specific set of computational resources.

---

### Official Review · Reviewer_YiSs · 2024-11-04

**Soundness:** 2
**Presentation:** 2
**Contribution:** 2
**Rating:** 3
**Confidence:** 2

**Summary:**

This paper addresses the significant challenge of mode collapse in time series generative models. The authors introduce a novel definition of mode collapse tailored specifically to time series data and propose a new metric, DMD-GEN, for its quantitative evaluation. DMD-GEN leverages Dynamic Mode Decomposition (DMD) to identify coherent spatio-temporal patterns and uses Optimal Transport to quantify discrepancies between real and generated data. The metric is tested across various recent time series generative models. Experimental results demonstrate DMD-GEN's effectiveness in evaluating the time series generative models.

**Strengths:**

- The paper studies the critical and underexplored problem of evaluating mode collapse in time series generative models. It also highlights the absence of specific metrics designed to address the dynamic characteristics of time series data.
- The proposed metric DMD-GEN is novel to the best of my knowledge. Furthermore, it provides valuable insights into the dynamics of generated time series, allowing for a deeper analysis in future research on time series modeling.
- DMD-GEN is computationally efficient as no additional training is required, which is very practical in real-time applications.

**Weaknesses:**

- The proposed metric effectively demonstrates that Diffusion-TS captures the target time series without mode collapse. However, the paper lacks a comparative analysis with models that are prone to mode collapse. It’s essential to provide such comparisons to verify the metric’s capability to discern between generative models that do and do not suffer from mode collapse.
- The robustness across different types of time series data is not thoroughly discussed, particularly for non-stationary time series with a low signal-to-noise ratio given the results on the stock dataset (please refer to Question 1 for a detailed concern).
- The presentation of results in figures and tables could be more self-contained. Specifically, the captions of Figures 1 and 2 should briefly explain the depicted results. Additionally, Table 1 should clearly indicate whether higher or lower values represent better performance.

**Questions:**

1. The differences between epoch 0 and the final epoch are significant in the comparisons of DMD eigenvalues of the original and generated time series across various datasets, with an exception noted in the stock dataset. Specifically, in Figure 2, the generated samples closely match the actual data from the start at epoch 0, with a slight improvement after the final epoch. Does it suggest that the DMD-GEN metric might not be as effective for highly non-stationary time series with low signal-to-noise ratio like stocks, compared to other types of data?
2. Do the results on the stock dataset in Figure 2 differ from Figure 6?
3. I am interested in exploring how the proposed DMD-GEN metric performs when applied to time series generated using traditional time series generation methods, e.g., Bootstrap. Could you extend your experiments to include this technique and provide the corresponding results?
4. Can you discuss the limitations of the DMD-GEN metric? Are there particular characteristics of time series datasets that might limit DMD-GEN's applicability, based on your current insight?

---

> ### Comment · Reviewer_YiSs · 2024-11-25
>
> As the rebuttal phase is approaching its conclusion, I would like to remind the authors that my initial rating was predicated on the expectation that you would address my concerns regarding the applicability of your approach for detecting model collapse. However, I have not yet received any responses to my questions. Please provide your responses by the end of the discussion period. If my concerns remain unaddressed, I will lower my final rating.

---

> ### Author Response · Authors · 2024-11-26
> **Response to Reviewer YiSs (1/2)**
>
> We thank the reviewer for their thoughtful comments and insightful questions. Below, we address the raised questions and weaknesses point-by-point:
>
> ***Response to Question 1: Effectiveness of DMD-GEN in Non-Stationary Datasets***
>
> Thank you for your insightful observation regarding the application of the DMD-GEN metric to highly non-stationary time series with low signal-to-noise ratios, such as stock market data.
>
> As outlined in our paper, the DMD-GEN metric is based on the eigenvectors (eigenmodes) corresponding to the largest absolute eigenvalues. These eigenmodes encode the most significant dynamic patterns in the data and are more informative than the eigenvalues alone because they capture the spatial structures of the underlying dynamics and inherently encapsulate information about the eigenvalues themselves.
>
> To clarify, the DMD eigenvalues are not explicitly used in the computation of the DMD-GEN metric (please refer to Algorithm 1). The purpose of Figures 1 and 2 is to demonstrate the informativeness of DMD eigenvalues in visualizing dynamic behaviors such as oscillations, exponential growth or decay. This visualization illustrates how well the generative models capture meaningful dynamic behaviors in time series data.
>
> Regarding your observation about the stock dataset, the generated samples closely matching the actual data from the start at epoch 0 may be attributed to the characteristics of stock market data. Highly non-stationary time series with low signal-to-noise ratios often exhibit less pronounced dominant dynamic patterns. As a result, the most significant eigenmodes are less distinct, leading to smaller observable differences between the initial and final epochs in terms of DMD eigenvalues.
>
> This does not suggest that the DMD-GEN metric is less effective for such data. Instead, it indicates that our proposed metric accurately reflects the inherent complexities and subtleties in the dynamics of highly non-stationary and noisy datasets. The slight improvements observed after training demonstrate that the model is still learning and refining its representation of the data's underlying dynamics, even if the changes are less dramatic compared to other datasets.
>
> While the DMD-GEN metric may show more subtle differences between epochs for datasets like stocks, it remains effective in capturing and evaluating the dynamic patterns. We appreciate your attention to this detail, as it highlights the versatility and robustness of the DMD-GEN metric across different types of time series data.
>
> ***Response to Question 2: Impact of Rank Selection in DMD Analysis - Clarifying Differences Between Figures 2 and 6***
>
> The primary difference between these figures lies in the rank selection used during the DMD process. In Figure 6, we employed a rank of 2 for the DMD decomposition. This choice was intentional to focus on the lower-dimensional dynamics of the stock dataset, emphasizing the most dominant modes that capture the primary trends and patterns in the data.
>
> In contrast, Figure 2 uses a higher rank for the DMD decomposition. By using a higher rank, we included more modes in the analysis, which allowed us to explore the higher-dimensional dynamics and capture more subtle variations within the dataset.
>
> By varying the rank in the DMD analysis between Figures 2 and 6, we aimed to demonstrate how the selection of rank affects the ability to capture different levels of dynamical behavior in the data. The results differ because each figure highlights different aspects of the dataset's dynamics based on the chosen rank during decomposition.
>
> We hope this clarifies the distinctions between the figures and underscores the impact of rank selection in DMD analyses. We are currently revising the manuscript to better explain and further clarify this point. Thank you very much.

---

> > ### Comment · Reviewer_YiSs · 2024-11-26
> >
> > Thanks for your detailed responses and efforts in conducting supplementary experiments. Some of my concerns are well-clarified. I recognize that this paper tries to develop a metric for detecting model collapse in time series modeling, which is a significant and pressing need within the field. I really expected that your proposed approach would be applicable, even if limited to stationary time series, and thus, gave a positive initial rating.
> >
> > To ensure our follow-up discussions are on the same page, I highlight the concerns from my initial review that remain unaddressed, listed in order of importance:
> > - (Weakness 2) The comparison with models prone to mode collapse. This is critical to demonstrate the applicability of the proposed metric - detecting whether generative models do and do not suffer from mode collapse. I note that this issue has also been raised by another reviewer.
> > - (Question 3) The application for evaluating traditional time series generation methods, e.g., Bootstrap, which you mentioned is part of an ongoing experiment in your response.
> >
> > I will await the results of these ongoing experiments before making my final decision.

---

> ### Author Response · Authors · 2024-11-26
> **Response to Reviewer YiSs (2/2)**
>
> ***Response to Question 4: Limitations of DMD-GEN Metric***
> Thank you for your question regarding the limitations of the DMD-GEN metric. Two significant factors that might limit its applicability are:
>
> 1. ***Applicability to Short Time Series:*** DMD-GEN relies on DMD to extract dynamic modes from time series data. This process requires a sufficient length of data to accurately capture the underlying system dynamics. With very short time series, there may not be enough temporal information to identify meaningful patterns or modes. This limitation can lead to less reliable or insightful results when applying DMD-GEN to datasets with limited time points.
> 2. ***Irregular or Missing Data Points:*** DMD assumes that the time series data is uniformly sampled without gaps. Irregular sampling intervals or missing data points can disrupt the DMD process, leading to inaccurate mode extraction or failure to converge on a solution. Such issues can significantly affect the performance of DMD-GEN, as the metric depends on the accurate decomposition of the time series dynamics. In practice, datasets with irregularities require preprocessing steps like interpolation or imputation, which may introduce additional errors.
>
> These limitations highlight scenarios where DMD-GEN might not provide optimal results, specifically, with short or incomplete time series data. Future work could focus on adapting DMD-GEN to handle such challenges, potentially by incorporating methods that accommodate irregular sampling or enhance performance with limited data.

---

> ### Author Response · Authors · 2024-11-26
>
> Dear Reviewer, We apologize for the delay in our response. While we are awaiting the benchmark results you requested (Question 3), we have provided detailed answers to all your other questions. We appreciate your patience and look forward to your feedback.

---

### Meta-Review · Area_Chair_EV1K · 2024-12-15

**Metareview:**

The paper proposes DMD-GEN, a novel metric based on Dynamic Mode Decomposition (DMD) and Grassmannian geometry to evaluate mode collapse in time series generative models. The authors claim it captures temporal dynamics more effectively than static metrics. Strengths include the novelty of addressing mode collapse in time series data, computational efficiency without retraining, and the provision of interpretability by identifying collapsed modes. However, weaknesses raised by reviewers include limited empirical validation, lack of comparisons with models prone to mode collapse, insufficiently addressed noise robustness, and unclear presentation of results and theoretical contributions. Ultimately, the lack of strong empirical evidence and clarity led to a borderline rejection recommendation.

**Additional Comments On Reviewer Discussion:**

During the rebuttal period, reviewers raised concerns about the limited empirical validation of the DMD-GEN metric, lack of comparisons with models prone to mode collapse, unclear theoretical explanations, and insufficient testing on diverse datasets. They also questioned the robustness of the metric to noise and its scalability to larger datasets. The authors addressed some of these points by expanding their analysis, conducting additional comparisons (e.g., with MTopDiv), and clarifying the mathematical basis of their approach. However, critical concerns, such as comparisons with collapse-prone models and noise robustness, remained insufficiently addressed. Additionally, issues with clarity and presentation persisted despite revisions. These unresolved concerns weighed heavily in the final decision, contributing to a recommendation for rejection, as the empirical and theoretical contributions were deemed incomplete and not yet ready for publication.

---

### Decision · Program_Chairs · 2025-01-22

Reject